# Potential roles of neuronal nitric oxide synthase and the PTEN-induced kinase 1 (PINK1)/Parkin pathway for mitochondrial protein degradation in disuse-induced soleus muscle atrophy in adult rats

Munehiro Uda[1]*, Toshinori Yoshihara[2], Noriko Ichinoseki-Sekine[2,3], Takeshi Baba[4], Toshitada Yoshioka[1]

1 School of Nursing, Hirosaki Gakuin University, Hirosaki, Aomori, Japan, 2 Graduate School of Health and Sports Science, Juntendo University, Inzai, Chiba, Japan, 3 Faculty of Liberal Arts, The Open University of Japan, Chiba, Japan, 4 School of Medicine, Juntendo University, Inzai, Chiba, Japan

* uda@hirogaku-u.ac.jp, uda1977jp@yahoo.co.jp

## Abstract

Excessive nitric oxide (NO) production and mitochondrial dysfunction can activate protein degradation in disuse-induced skeletal muscle atrophy. However, the increase in NO production in atrophied muscles remains controversial. In addition, although several studies have investigated the PTEN-induced kinase 1 (PINK1)/Parkin pathway, a mitophagy pathway, in atrophied muscle, the involvement of this pathway in soleus muscle atrophy is unclear. In this study, we investigated the involvement of neuronal nitric oxide synthase (nNOS) and the PINK1/Parkin pathway in soleus muscle atrophy induced by 14 days of hindlimb unloading (HU) in adult rats. HU lowered the weight of the soleus muscles. nNOS expression showed an increase in atrophied soleus muscles. Although HU increased malondialdehyde as oxidative modification of the protein, it decreased 6-nitrotryptophan, a marker of protein nitration. Additionally, the nitrosocysteine content and S-nitrosylated Parkin were not altered, suggesting the absence of excessive nitrosative stress after HU. The expression of PINK1 and Parkin was also unchanged, whereas the expression of heat shock protein 70 (HSP70), which is required for Parkin activity, was reduced in atrophied soleus muscles. Moreover, we observed accumulation and reduced ubiquitination of high molecular weight mitofusin 2, which is a target of Parkin, in atrophied soleus muscles. These results indicate that excessive NO is not produced in atrophied soleus muscles despite nNOS accumulation, suggesting that excessive NO dose not mediate in soleus muscle atrophy at least after 14 days of HU. Furthermore, the PINK1/Parkin pathway may not play a role in mitophagy at this time point. In contrast, the activity of Parkin may be downregulated because of reduced HSP70 expression, which may contribute to attenuated degradation of target proteins in the atrophied soleus muscles after 14 days of HU. The present study provides new insights into the roles of nNOS and a protein degradation pathway in soleus muscle atrophy.

**Data Availability Statement:** All relevant data are within the manuscript and its Supporting Information files.

**Funding:** This work was supported by JSPS KAKENHI Grant Numbers JP26560402, JP19K11554 to MU.

**Competing interests:** The authors have declared that no competing interests exist.

## Introduction

Prolonged skeletal muscle inactivity such as hindlimb unloading in rodents results in skeletal muscle atrophy. Skeletal muscle atrophy occurs because of both a reduced rate of protein synthesis and an increased rate of degradation [1–3]. The protein synthesis is regulated by Akt/mechanistic target of rapamycin (mTOR) signaling pathway, which is suppressed by prolonged skeletal muscle inactivity [4]. In contrast, the protein degradation is regulated by ubiquitin-proteasome system [5], autophagy-lysosomal system [6,7], calpain [8,9], and caspase 3 [10]. Excessive oxidative stress [11], calcium overload [12], and the impairment of mitochondrial metabolic function and dynamics [13,14] trigger the activation of protein degradation systems in disuse-induced skeletal muscle atrophy. Neuronal nitric oxide synthase (nNOS) and/or nitric oxide (NO) were also reported to be involved in these systems [15]. However, the mechanisms of the involvement of nNOS and/or NO in these systems are unclear [11,16].

The increase or decrease in NO production in the disuse-induced atrophied muscle remains controversial. A previous study has indicated that nNOS and/or NO mediate 14 days of tail suspension-induced skeletal muscle atrophy by activating of muscle-specific RING finger protein 1 and muscle atrophy F-box 1/atorogen-1, which are E3 ubiquitin ligases [15]. This conclusion is based on the observation that both nNOS-null mice and wild type mice receiving an nNOS inhibitor showed a reduced degree of disuse-induced muscle atrophy [15]. Additionally, NO production and nNOS catalytic activity, have been reported to be increased in wild type tail suspended mice despite a decrease in nNOS expression [15]. The prevention of muscle atrophy by the inhibition of nNOS was also demonstrated in another study [17]. In contrast, other previous studies showed that the expression of nNOS and the production of NO is decreased in atrophied soleus muscles after 10 to 14 days of HU [18,19]. Therefore, nNOS has been shown to have paradoxical roles in disuse-induced skeletal muscle atrophy.

The activity of nNOS is regulated by post-translational modifications and by interactions with other proteins. Additionally, nNOS produces not only NO but also superoxide, a reactive oxygen species (ROS), in some cases. Furthermore, the ubiquitination of nNOS are affected by molecular chaperones. Phosphorylation of nNOS at Ser1412 has been reported to increases its activity [20]. In contrast, phosphorylation of nNOS at Ser847 results in decreases NO production and induced ROS production by nNOS [21,22]. Binding of calmodulin (CaM) and heat shock protein 90 (HSP90) to nNOS is also known to promote NO production [23–25], whereas nNOS produces ROS rather than NO in the context of low levels of HSP90 [26]. Ubiquitination of nNOS is also reduced in the absence of HSP70 [27].

PTEN-induced kinase 1 (PINK1)/Parkin signaling pathway mediates mitochondrial autophagy, namely mitophagy. The signaling pathway is also regulated by NO-related post-translational modifications and interactions with a molecular chaperone [28,29]. Although the mechanisms of the PINK1/Parkin-mediated mitophagy are yet not fully understood, a model revealing the pathway mechanisms has been proposed in a recent review article [30]. PINK1 accumulates in dysfunctional mitochondria and phosphorylates Parkin, an E3 ubiquitin ligase. The phosphorylated Parkin ubiquitinates mitochondrial outer membrane proteins, such as mitofusin 2 (MFN2) [31], and leads to the degradation of ubiquitinated proteins in the proteasome followed by the induction of mitophagy. The activity of Parkin is also regulated by S-nitrosylation, which is the addition of NO to cysteine residues in proteins [28]. Nitrosative stress conditions, which result from the production of excessive reactive nitrogen species (RNS), inhibits Parkin activity through excessive S-nitrosylation [28]. In contrast, Parkin activity is enhanced by binding to HSP70 [29]. The involvement of the PINK1/Parkin pathway in skeletal muscle atrophy has been investigated using gastrocnemius (GAS) and tibialis anterior (TA) muscles after skeletal muscle inactivity [32–36]. The expression of PINK1 and/or Parkin

in GAS, which is composed of both slow- and fast-twitch fibers, was unchanged or decreased by disuse-induced skeletal muscle atrophy [33–36]. In contrast, their expression in TA, which is composed of predominantly fast-twitch fibers, was increased or remained unchanged after skeletal muscle inactivity [32,36]. However, the role of this pathway in the atrophy of the slow-twitch predominant soleus muscle is unclear. Thus, it is interesting to determine whether the PINK1/Parkin pathway is involved in disuse-induced soleus muscle atrophy together with the potential changes in the production of NO.

The role of nNOS and/or NO in disuse-induced soleus muscle atrophy remains unclear. In addition, the activation of the PINK1/Parkin pathway in atrophied soleus muscles requires further analysis. Therefore, we examined the involvement of nNOS and the PINK1/Parkin pathway in disuse-induced soleus muscle atrophy. Rat hindlimbs were unloaded for 14 days and soleus muscles were analyzed. Our data showed that NO production in atrophied soleus muscles does not increase even when activated nNOS is elevated. In addition, we show that the PINK1/Parkin pathway may not be active in atrophied soleus muscles. Our results also indicate that the basal activity of Parkin may be downregulated in the atrophied soleus muscles.

## Materials and methods

### Animals

Animal care was performed in accordance with the Guidelines for Proper Conduct of Animal Experiments by the Science Council of Japan. The institutional animal care and use committee of Juntendo University approved all experimental protocols (approval number: H27-05). Fifteen-week-old male Fischer F344/N rats (n = 12) were obtained from Japan SLC, Inc. (Shizuoka, Japan). The rats were housed under a 12:12-h light-dark cycles in a controlled-environment room (23 ± 1˚C, 55 ± 5% relative humidity), and were provided with food and water ad libitum. One week after arrival from the vendor, the rats were randomly assigned to control (CON, n = 6) and hindlimb unloading (HU, n = 6) groups.

### Hindlimb unloading

Each of the rats in the HU group was exposed to tail suspension for 14 days using the method described by Yoshihara et al. [9]. We chose this time point because, in previous studies, NO production and nNOS expression were evaluated in isolated muscles after 14 days of HU [15,18]. Briefly, a tail cast was applied to each rat, leaving the distal one-third of the tail free to allow for proper thermoregulation. The tail cast was attached to a hook on the ceiling of the cage, and the height of the hook was adjusted at an inclination of approximately 35˚ in a head-down orientation. The rat was free to move around the cage on its front feet. Rats were checked daily for signs of tail lesions or discoloration.

### Muscle preparation

Rats from the CON and HU groups were deeply anesthetized with pentobarbital sodium. Once the rats became completely unresponsive to stimulation, the soleus muscles were removed and frozen in liquid nitrogen. Then, the rats were euthanized by exsanguination. Protein extraction and biochemical analysis were performed in a manner similar to that reported in our previous studies [37,38]. Briefly, the frozen soleus muscles were homogenized in a lysis buffer containing 40 mM Tris, 8 M urea, 4% CHAPS, 65 mM dithiothreitol (DTT), 1 mM EDTA, 20 mM N-ethylmaleimide (NEM), and cOmplete Protease Inhibitor (Roche Applied Science, Basel, Switzerland). The homogenates were then centrifuged at 15000 ×g for 15 min at 4˚C, and the middle layer, containing the proteins, was carefully collected.

## Western blot analysis

Equal amounts of proteins (30–200 μg/lane) were loaded onto polyacrylamide gels, and the proteins were separated using sodium dodecyl sulfate gel electrophoresis (SDS-PAGE). Proteins in the gels were then transferred onto a PVDF membranes (Immobilon-P, 0.45 μm; Millipore-Merk, Darmstadt, Germany). Non-specific binding sites were blocked for 1–2 h at room temperature with 5% bovine serum albumin (BSA) in Tris-buffer saline containing Tween 20, pH 7.6 (TBST). The membranes were incubated overnight to 2 days at 4˚C or 1 h at room temperature with primary antibodies diluted in Can Get Signal Immunoreaction Enhancer Solution 1 (TOYOBO CO., LTD) or TBST containing 3% bovine serum albumin (BSA). The membranes then were incubated for 1 h at room temperature with alkaline phosphatase-conjugated secondary antibodies diluted in Can Get signal solution 2 or TBST with 1% BSA after washing with TBST. The signals were then visualized using Immunstar-AP substrate (Bio-Rad, Hercules, CA, USA), and the membranes were exposed to Hyperfilm ECL (GE Healthcare life science). The primary antibodies used in the present study were as follows: rabbit anti-nNOS monoclonal antibody (C7D7, #4231, Cell Signaling Technology, diluted 1:1000), rabbit anti-nNOS (phospho S1417) polyclonal antibody (ab5583, Abcam, diluted 1:2000), rabbit anti-nNOS (phospho S847) polyclonal antibody (ab16650, Abcam, diluted 1:4000), rabbit anti-calmodulin polyclonal antibody (#4830, Cell Signaling Technology, diluted 1:1000), rabbit anti-HSP90 polyclonal antibody (ab13495, Abcam, diluted 1:1000), mouse anti-HSP70 monoclonal antibody (C92F3A-5, ADI-SPA-810, Enzo Life Sciences, diluted 1:1000), mouse anti-malondialdehyde (MDA) monoclonal antibody (MMD-030n, Japan Institute for the Control of Aging, NIKKEN SEIL CO., Ltd, diluted 1:200), mouse anti-6-nitrotryptophan (6-NO$_2$Trp) monoclonal antibody (Japan Institute for the Control of Aging, NIKKEN SEIL CO., Ltd, 1μg/ml), mouse anti-nitrotyrosine (3-NT) monoclonal antibody (39B6, SC-32757, Santa Cruz Biotechnology, Inc. diluted 1:1000), mouse anti-nitrosocystein monoclonal antibody (HY8E12, ab94930, abcam, diluted 1:1000), rabbit anti-PINK1 polyclonal antibody (BC100-494, Novus Biologicals, diluted 1:2000), mouse anti-Parkin monoclonal antibody (PRK8, ab77924, Abcam, diluted 1:1000), mouse anti-MFN2 monoclonal antibody (6A8, ab56889, Abcam, diluted 1:1000), and mouse anti-mono- and polyubiquitinylated conjugates monoclonal antibody (FK2, BML-PW8810-0100, Enzo, diluted 1:1000). The secondary antibodies used were alkaline phosphatase (AP)-conjugated donkey anti-mouse IgG (715-035-150, Jackson ImmunoResearch Laboratories, West Grove, PA, USA, diluted 1:100000) and AP-conjugated donkey anti-rabbit IgG (715-035-150, Jackson ImmunoResearch Laboratories, West Grove, PA, USA, diluted 1:100000).

SYPRO Ruby was used for protein detection. The membranes were stained with SYPRO Ruby Protein Blot Stain (Life Technologies, Carlsbad, CA, USA, and Lonza Rockland, Inc., Rockland, ME, USA) immediately after the protein transfer to membranes, or after the antibodies were stripped from the membranes. The signals were visualized using an LED Transilluminator (LED470-TR60W, MeCan image, Saitama, Japan) and the images were obtained using a digital camera (Canon, EOS Kiss X7).

## Immunoprecipitation

Parkin immunoprecipitation (IP) was performed by exchanging the 8 M urea buffer to IP buffer (pH 7.5) containing 10 mM Tris-HCl, 150 mM NaCl, 1% Nonidet P-40, 20 mM NEM, and protease inhibitor cocktail containing EDTA (cOmplete ULTRA Tablets, Mini, EASYpack, Merck) using acetone precipitation. In contrast, the buffer was not changed from the 8 M urea buffer to the IP buffer for MFN2 IP as insoluble protein naturally occurred during Parkin IP. Therefore, the protein sample was directly added to the IP buffer, which prevented the

formation of insoluble protein in the IP buffer after adding the protein sample. In addition, MFN2 IP was performed using 6 samples from the CON group and 4 samples from the HU group because the amount of 2 samples in the HU group was insufficient. Antibody-bound beads (100 μl of SureBeads Protein A, Bio-Rad laboratories) were washed with PBS containing 0.1% Tween 20 (PBST). The beads were suspended in PBST containing either 2 μg of anti-Parkin antibody (PRK8) or anti-MFN2 antibody (6A8) and rotated for 10 min at room temperature. The beads were then magnetized and the supernatants were removed. Subsequently, IP buffer containing 500 μg of protein was added to the beads and rotated overnight at 4˚C. The antigen-bound beads were magnetized and washed three times with IP buffer without protease inhibitors. The bound antigens were solubilized with sample buffer and incubated for 10 min at 70˚C. The protein samples were then loaded onto polyacrylamide gels, and western blotting was performed to detect immunoreactive bands.

### Image analysis

The film images were scanned for densitometric analysis. The western blot analysis and SYPRO Ruby staining signals were quantified using ImageJ software.

### Statistical analysis

Statistical analysis was carried out using IBM SPSS Statistics software, version 24. All data are presented as means ± standard error (SE). Normality was confirmed using the Shapiro-Wilk test. Based on the distribution of the data, either independent samples t-test or Mann-Whitney U-test were used to analyze the differences between the CON and HU groups. Statistical significance was set at $p < 0.05$.

## Results

### Body weight, soleus muscle weight, and relative soleus muscle weight

The body weight, soleus muscle weight, and relative soleus muscle weight in rats from the CON and HU groups are shown in Fig 1. The body weight in the HU group was significantly lower after 14 days of HU (22.1% lower in HU vs. CON, $p < 0.001$, Fig 1A). The soleus muscle weight in the HU group was also significantly lower than that seen in the CON group (39.2%

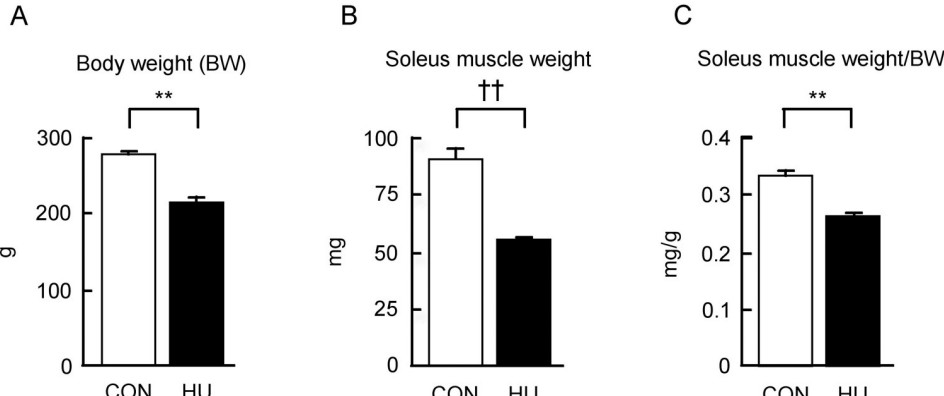

**Fig 1. Changes in the body weight, soleus muscle weight, and relative soleus muscle weight after hindlimb unloading.** (A-C) Changes in body weight (BW), soleus muscle weight, and relative soleus muscle weight to BW are indicated in A, B, and C, respectively. CON, control. HU, hindlimb unloading. Data are expressed as means ± SE. \*\*significantly different from CON ($p < 0.01$). ††significantly different from CON ($p < 0.01$, Mann-Whitney U-test).

lower in HU vs. CON, p = 0.004, Mann-Whitney U-test, Fig 1B). Additionally, the relative soleus weight in the HU group was lower than that in the CON group (21.6% lower in HU vs. CON, p < 0.001, Fig 1C). These results indicate that soleus muscle atrophy was induced by 14 days of HU.

## nNOS and phosphorylated nNOS expression

The effect of HU on the expression of nNOS and phosphorylated nNOS (p-nNOS) was evaluated. nNOS have several splice variants including nNOSμ, which is mainly expressed in the skeletal muscle [39,40]. nNOSμ is the form with additional 34 amino acids inserted between the residues Lys839 and Ser840 of nNOSα [39]. Therefore, the phosphorylation sites at Ser1412 and Ser847 of nNOSα correspond to those at Ser1446 and Ser881of nNOSμ, respectively. In the present study, p-nNOS at Ser1412 and p-nNOS at Ser847 are referred to as p-nNOS at Ser1446 and p-nNOS at Ser881, respectively. The results are shown in Fig 2. Strong immunoreactivity of nNOS was detected in the HU group but was not detected in the CON group (Fig 2A). The immunoreactivity of p-nNOS at Ser1446 was also stronger in the HU group than that in the CON group (Fig 2A). Moreover, the immunoreactivity of p-nNOS at Ser881 immunoreactivity was detected in both groups at similar levels (Fig 2A). The intensities of the bands were normalized by total protein levels (Fig 2B), and the values were expressed as fold change relative to the CON group as shown in Fig 2C–2E. nNOS expression in the total protein was significantly higher (2.5-fold) in the HU group than that in the CON group (p < 0.001, Fig 2C). Additionally, the expression of p-nNOS at Ser1446 in the total protein in the HU group was also significantly higher (2.3-fold) than that in the CON group (p = 0.02, Fig 2D). However, no difference was observed between the expression of p-nNOS at Ser881 in the total protein between the CON and HU groups (p = 0.378, Fig 2E). Additionally, the values of the p-nNOS expression levels normalized to the nNOS expression levels were calculated, and the values of p-nNOS at Ser1446 and Ser881 were indicated as fold change relative to the CON group as shown in Fig 2F and 2G, respectively. The expression levels of p-nNOS at Ser1446 in the nNOS expression levels were not significantly different between the CON and HU groups (p = 0.845, Fig 2F). The expression levels of p-nNOS at Ser881 in the nNOS expression levels were significantly lower (0.5-fold) in the HU group than that in the CON group (p = 0.001, Fig 2G). These results indicate nNOS expression and p-nNOS at Ser1446 were increased in response to 14 days of HU-induced atrophy in soleus muscles.

## Expression of nNOS-interacting proteins

nNOS activity is regulated by the binding of CaM and HSP90 [23–26]. In addition, nNOS also binds to HSP70 [41]. Thus, the effects of HU on the content of CaM, HSP90, and HSP70 were also evaluated in this study. Fig 3 shows the immunoreactivities of CaM, HSP90, and HSP70. The immunoreactivity of CaM appeared to be slightly different between the CON and HU groups (Fig 3A), whereas the immunoreactivities of both HSP90 and HSP70 in the HU group were weaker than that in the CON group (Fig 3A). These immunoreactivities were normalized to the total proteins (Fig 3B), and the values were expressed as fold change relative to the CON group as shown in Fig 3C–3E. Although CaM expression in the HU group showed a tendency to decrease as compared to the CON group, statistical significance was not observed (p = 0.125, Fig 3C). In contrast, the expression of HSP90 (0.5-fold, p = 0.004, Fig 3D) and HSP70 (0.4-fold, p = 0.036, Fig 3E) in the HU group was significantly lower than that in the CON group.

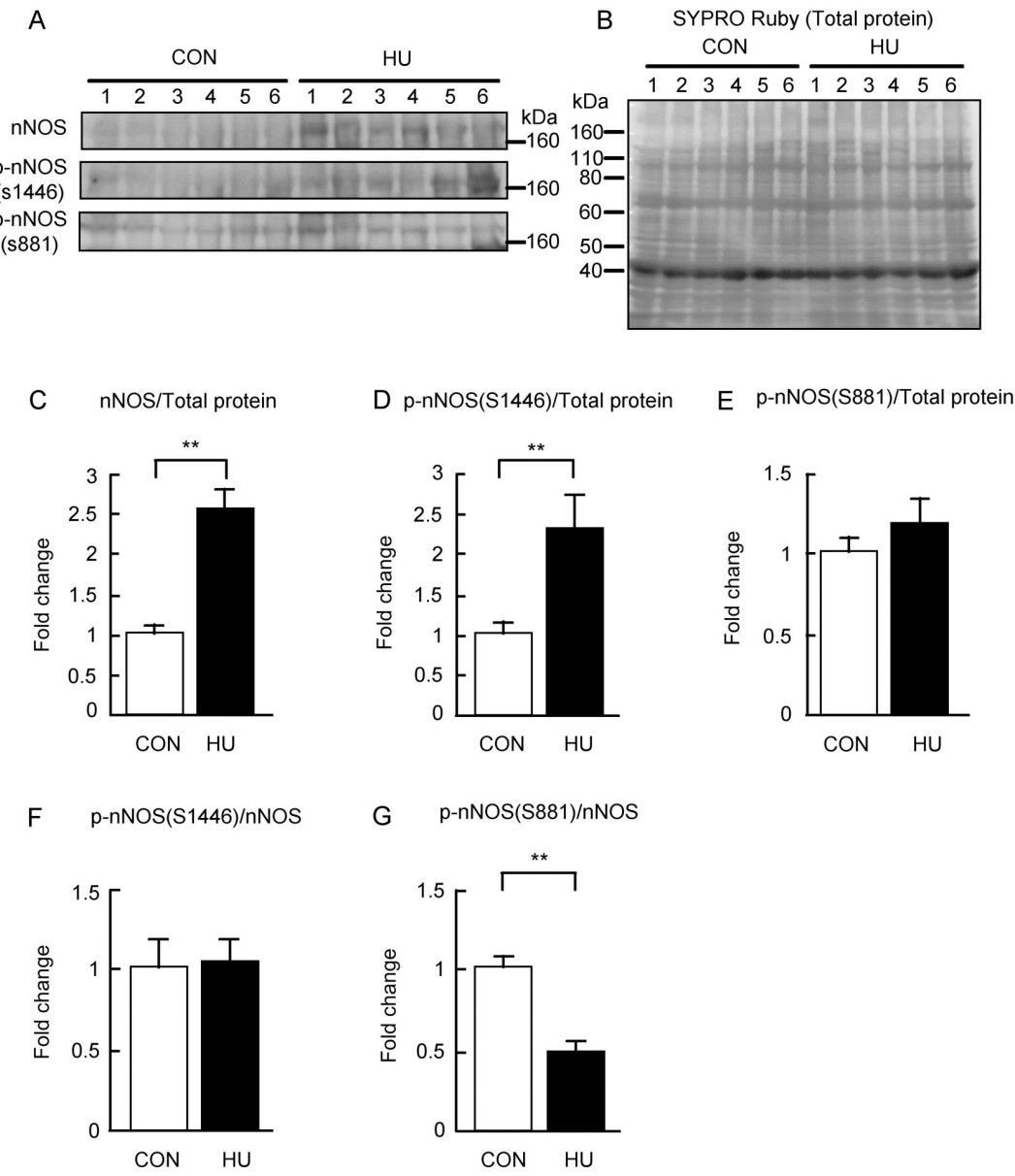

**Fig 2. Changes in the expression levels of nNOS and phosphorylated nNOS after hindlimb unloading.** (A) Western blots showing immunoreactivities of nNOS, phosphorylated nNOS at S1446, and phosphorylated nNOS at S881. (B) Total membrane protein detected by SYPRO Ruby staining. (C-E) Comparisons of nNOS, phosphorylated nNOS at S1446, and phosphorylated nNOS at S881 normalized to total protein between groups are indicated in C, D, and E, respectively. (F, G) Comparisons of phosphorylated nNOS at S1446 and phosphorylated nNOS at S881 normalized to nNOS expression levels between groups are indicated in F and G, respectively. CON, control. HU, hindlimb unloading. Data are expressed as means ± SE. **significantly different from CON (p < 0.01).

## Oxidative and nitrative modification

Previous studies showed that oxidative modifications of protein are increased in atrophied skeletal muscle [13,42]. It has also been reported that protein nitration does not change or is decreased in atrophied skeletal muscle [43,44]. Therefore, the effects of HU on the levels of oxidative and nitrative modifications were investigated in this study using 6-$NO_2$Trp along with 3-NT as markers of protein nitration. 6-$NO_2$Trp is induced by RNS such as peroxynitrite,

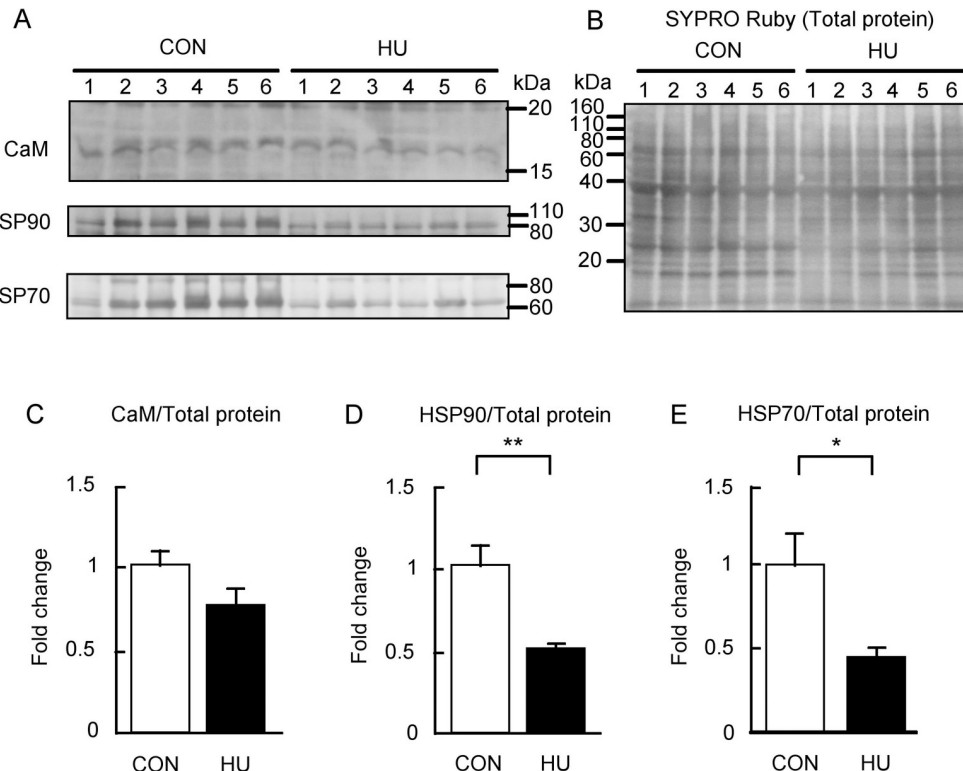

**Fig 3. Changes in the expression of nNOS interacting proteins after hindlimb unloading.** (A) Western blots showing immunoreactivities of calmodulin (CaM), heat shock protein 90 (HSP90), and heat shock protein 70 (HSP70). (B) Total membrane protein detected by SYPRO Ruby staining. (C-E) Comparisons of CaM, HSP90, and HSP70 normalized to total protein between groups are indicated in C, D, and E, respectively. CON, control. HU, hindlimb unloading. Data are expressed as means ± SE. **significantly different from CON (p < 0.01). *significantly different from CON (p < 0.05).

which is produced by the reaction of NO and superoxide [45,46]. Previous studies demonstrated that 6-NO$_2$Trp is a marker for oxidative and nitrative stress [47–49]. The results of western blotting and semi-quantitative analysis are shown in Fig 4. Strongly stained bands for MDA were observed in the HU group (Fig 4A), whereas the immunoreactivity of 6-NO$_2$Trp was reduced in the HU group as compared to in the CON group (Fig 4B). Similarly, the immunoreactivity of 3-NT was weaker in the HU group than that in the CON group (Fig 4C). The value of each immunoreactive band was normalized to the total protein (Fig 4D) and indicated as fold change of the HU group relative to the CON group (Fig 4E–4G). The MDA content was significantly greater (1.4-fold) in the HU group than that in the CON group (p = 0.004, Fig 4E) similar to the previous observation [42], whereas the 6-NO$_2$Trp content was significantly lower in the HU group (0.5-fold, p = 0.003, Fig 4F). 3-NT content tended to be lower with HU (p = 0.176, Fig 4G). Thus, these results indicate that while HU increased oxidative modification of the protein, it decreased protein nitration in atrophied soleus muscles.

## Nitrosocysteine content and NO-related Parkin modification

Previous studies have reported that S-nitrosylation may be affected by changes in NO production [28]. Thus, the nitrosocysteine content in total protein was evaluated using a specific antibody (Fig 5). The immunoreactivity of nitrosocysteine, the SYPRO Ruby stained membrane, and the nitrosocysteine level in total protein are shown in Fig 5A, 5B and 5D, respectively. No

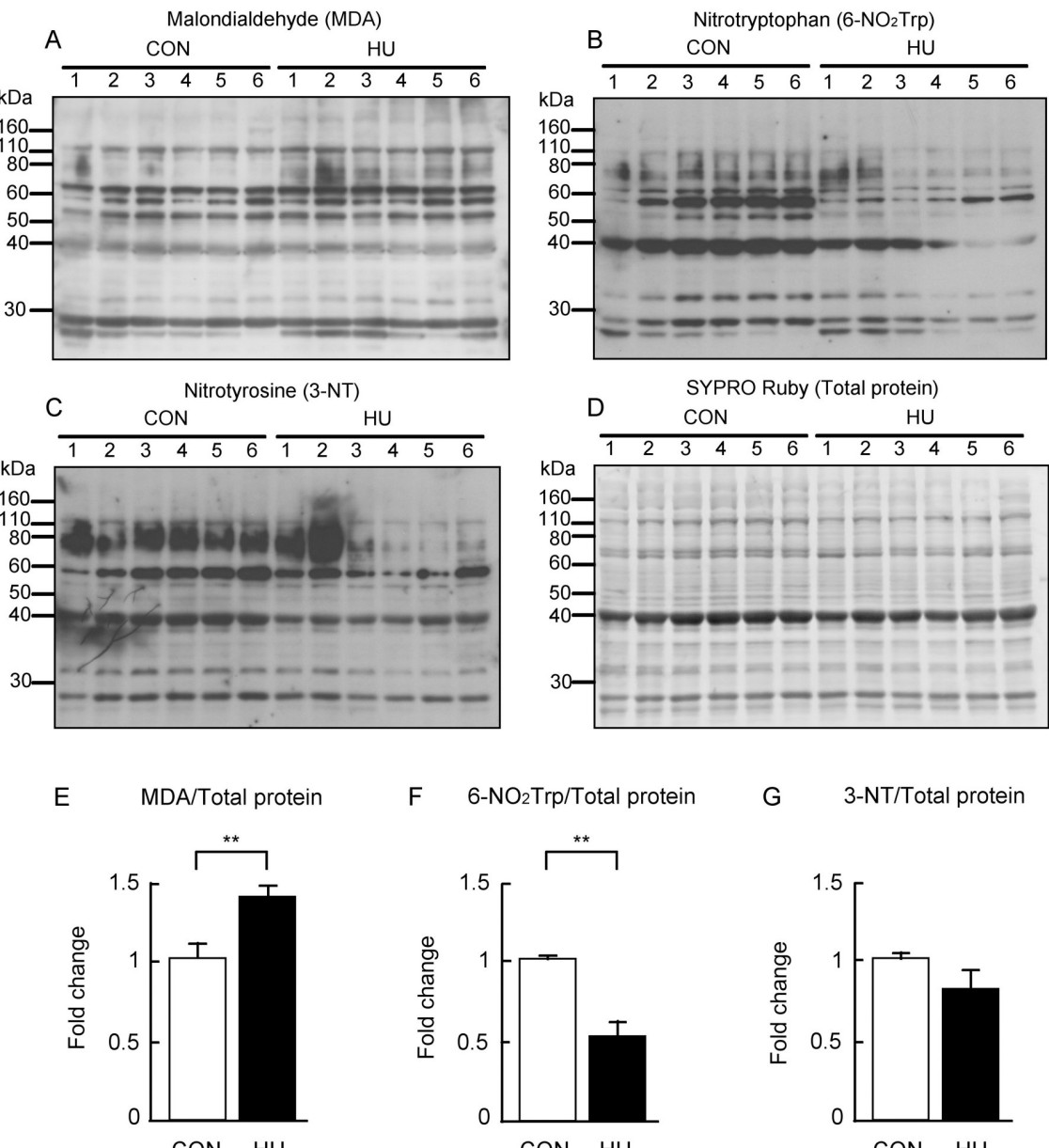

**Fig 4. Changes in the protein oxidation and nitration after hindlimb unloading.** Western blots showing immunoreactivities of (A) malondialdehyde (MDA), (B) 6-nitrotryptophan (6-NO$_2$Trp), and (C) 3-nitrotyrosine (3-NT). Each immunoreactivity level was normalized to (D) total protein. (E-G) Comparison of MDA, 6-NO$_2$Trp, and 3-NT contents between groups are indicated in E, F, and G, respectively. CON, control. HU, hindlimb unloading. Data are expressed as means ± SE. **significantly different from CON ($p < 0.01$).

difference was found in the nitrosocysteine levels between groups (p = 0.337, Mann-Whitney U-test, Fig 5D). In addition, the S-nitrosylated and nitrated Parkin levels were evaluated using IP, as both S-nitrosylation and nitration of Parkin are also affected by NO and peroxynitrite production [50]. Parkin has been reported to have multiple isoforms and two major bands have been observed at 50 kDa and 44 kDa in rats and mice brains using western blotting [51,52]. In this study, approximately 40 kDa Parkin was only precipitated by IP (Fig 5C). Therefore, the S-nitrosylation and tryptophan nitration of only approximately 40 kDa Parkin

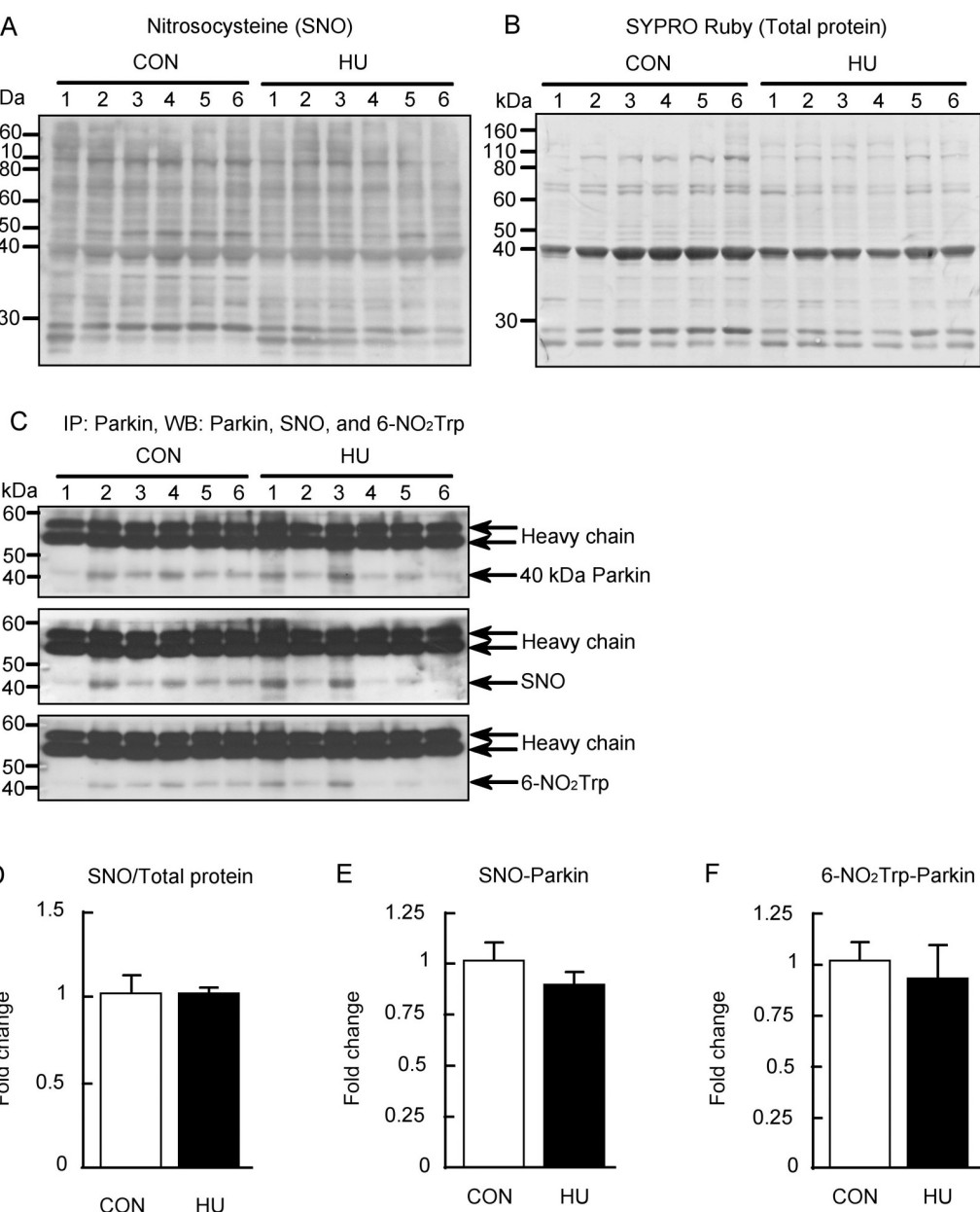

**Fig 5. S-nitrosylation and tryptophan nitration of Parkin after hindlimb unloading.** (A) Nitrosocysteine (SNO) detected by western blot analysis. (B) Total membrane protein detected by SYPRO Ruby staining. (C) Immunoprecipitation performed using anti-Parkin antibody, and Parkin, SNO, and 6-nitrotryptophan (6-NO$_2$Trp) immunoreactivities were detected by western blot analysis. (D) Comparisons of SNO immunoreactivity normalized to total protein between groups. (E, F) Comparisons of S-nitrosylated (SNO)-Parkin and tryptophan nitrated (6-NO$_2$Trp)-Parkin normalized to Parkin expression levels between groups are indicated in E and F, respectively. CON, control. HU, hindlimb unloading. Data are expressed as means ± SE.

was evaluated. The band intensities of nitrosocysteine and 6-NO$_2$Trp at approximately 40 kDa Parkin were normalized to the immunoprecipitated Parkin levels, and the values of fold change relative to the CON group are shown in Fig 5E and 5F. The S-nitrosylation of Parkin was slightly lower in the HU group, although not significantly (p = 0.27, Fig 5E). Tryptophan nitration of Parkin was also demonstrated to be unaffected by HU (p = 0.634, Fig 5F). These results indicate that the S-nitrosylation and nitration of Parkin were unchanged by HU.

## Expressions of PINK1, Parkin, and MFN2

The PINK1/Parkin pathway and MFN2 as a target protein of Parkin were further studied (Fig 6). The immunoreactivities of PINK1, 50 kDa Parkin, approximately 40 kDa Parkin, and MFN2, as well as the SYPRO Ruby stained membranes, are shown in Fig 6A and 6B. No difference was observed in the immunoreactivities of PINK1 and approximately 40 kDa Parkin in the HU group as compared to in the CON group. Immunoreactivity of 50 kDa Parkin, particularly in the HU group, showed individual differences that were not observed in other immunostaining experiments. In addition, MFN2 immunoreactivity was observed at approximately 80 kDa, which is the predicted molecular weight. However, MFN2 immunoreactivities were also observed at 110 kDa and between 110 and 160 kDa similar to reported in previous studies [31,53]. Compared to the CON group, the HU group showed a longitudinally wide MFN2 immunoreactivity between 110 and 160 kDa (Fig 6A). The band intensities were normalized to those of the total proteins (Fig 6B) and the values were expressed as fold change relative to those in the CON group as shown in Fig 6C–6F. PINK1 expression did not differ between groups (p = 0.337, Mann-Whitney U-test, Fig 6C). The expression of 50 kDa Parkin tended to be higher in the HU group as compared to that in the CON group (p = 0.17, Fig 6D), whereas the expression of approximately 40 kDa Parkin in the HU group was slightly lower than that in the CON group (p = 0.208, Fig 6E). Statistical significance was not observed between groups for both molecular weights of Parkin. Similarly, HU did not affect the expression of 80 kDa (p = 0.071) and 110 kDa (p = 0.183) MFN2, while the expression of MFN2 between 110 and 160 kDa tended to be higher in the HU group (p = 0.423, Mann-Whitney U-test, Fig 6F). Total MFN2 expression in the HU group was slightly higher than that in the CON group (p = 0.522, Mann-Whitney U-test, Fig 6F). In addition, MFN2 individual band intensities were normalized to the total MFN2 expression level and the values are expressed as percentages in Fig 6G. Most of MFN2 was observed between 110 and 160 kDa in both the CON and HU groups. In addition, a significant reduction was observed in the percentage expression of 80 kDa (15% in CON vs. 10% in HU, p = 0.012) and 110 kDa (23% in CON vs. 17% in HU, p = 0.002) MFN2 in the HU group as compared to in the CON group (Fig 6G). In contrast, a significant increase was observed in the percentage expression of MFN2 between 110 and 160 kDa (56% in CON vs. 64% in HU) in the HU group as compared to in the CON group (p = 0.004, Fig 6G). These results indicated that HU did not alter the expression of PINK1 and Parkin, whereas it may induce the accumulation of high molecular weight MFN2 in the atrophied soleus muscles.

We additionally confirmed whether the high molecular weight MFN2 could be detected by IP. Fig 7A shows the immunoreactivities of MFN2 and ubiquitin after MFN2 IP. Immunoreactivity of MFN2 was observed between 110 and 160 kDa in both groups, which was similar to the results obtained for western blot analysis of MFN2 (Fig 6A). This suggests that the high molecular weight MFN2 is present in the atrophied soleus muscles and proves MFN2 immunoreactivity between 110 and 160 kDa (Fig 6A). In addition, the MFN2 immunoreactivity between 110 and 160 kDa in the HU group was higher than that in the CON group. This result agreed with the percentage of MFN2 in the CON and HU groups (Fig 6G). The accumulation of high molecular weight MFN2 in the HU group may have resulted from the reduced ubiquitination of MFN2 in atrophied soleus muscles. Thus, we evaluated the ubiquitination of high molecular weight MFN2 (Fig 7). In contrast to MFN2, ubiquitin immunoreactivity did not differ between groups or was slightly weaker in the HU group. Ubiquitin immunoreactivities between 110 and 160 kDa were normalized to the immunoprecipitated MFN2 levels between 110 and 160 kDa, and the values of fold change relative to the CON group is shown in Fig 7B. The ubiquitination of high molecular weight MFN2 was significantly lower (0.16-fold) in the

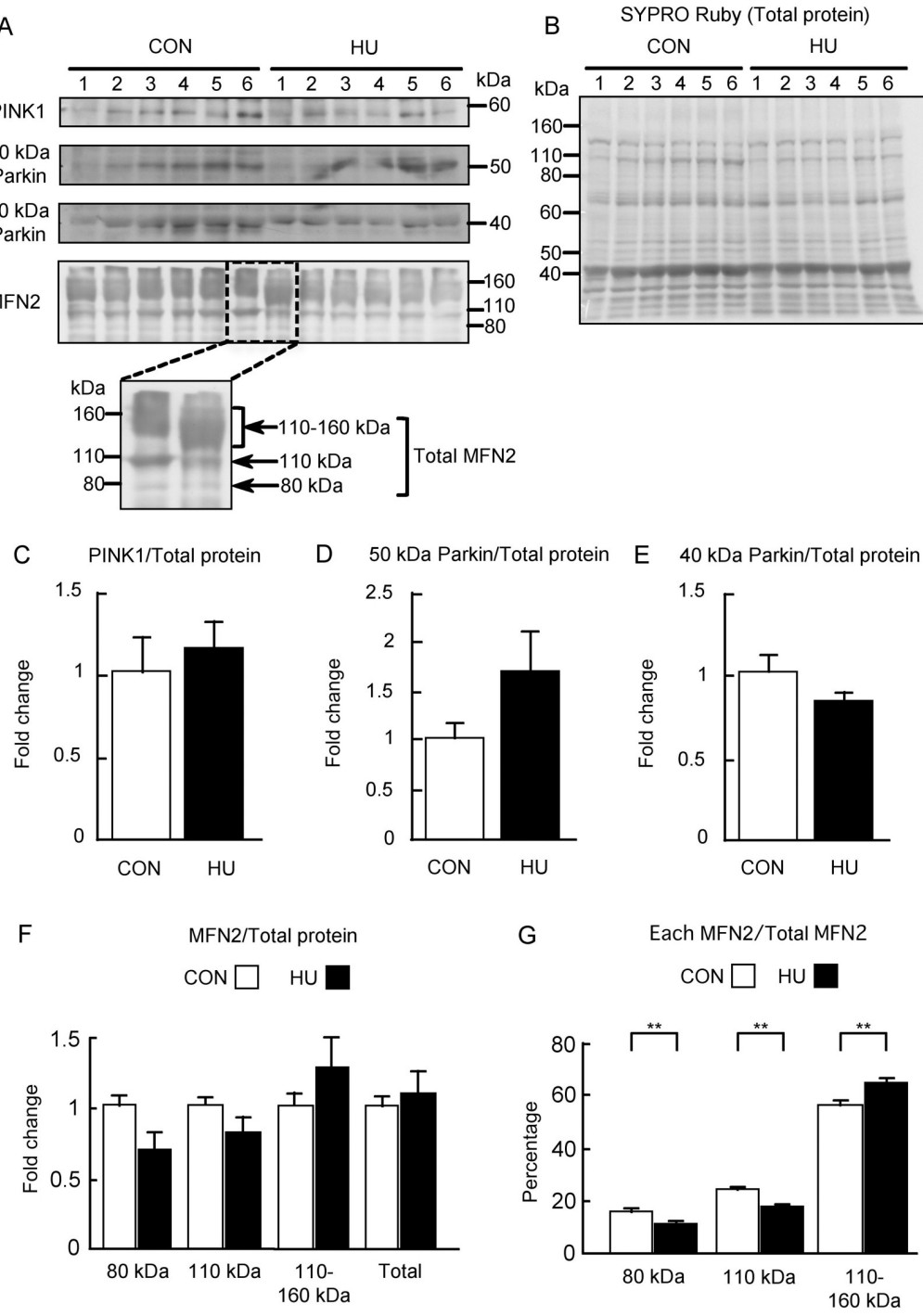

**Fig 6. Expression of PINK1, Parkin, and MFN2 after hindlimb unloading.** (A) Western blots of PINK1, Parkin, and MFN2 immunoreactivities. (B) Total membrane protein detected by SYPRO Ruby staining. (C-F) Comparisons of the immunoreactivities of PINK1, 50 kDa Parkin, approximately 40 kDa Parkin, and MFN2 expression normalized to total protein between groups are indicated in C, D, E, and F, respectively. (G) Comparison of relative MFN2 level calculated from each MFN2 band intensity normalized to total MFN2 is expressed in G as a percentage. CON, control. HU, hindlimb unloading. Data are expressed as means ± SE. **significantly different from CON ($p < 0.01$).

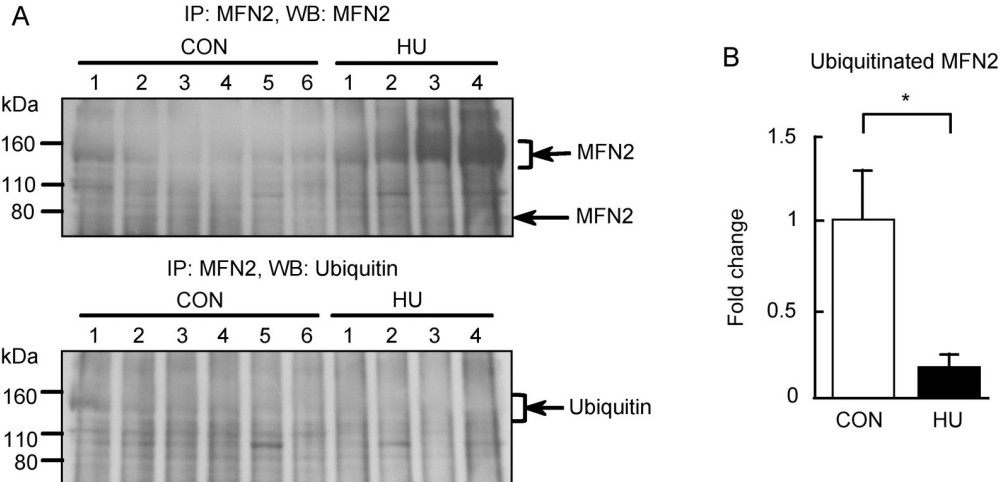

**Fig 7. Ubiquitination of MFN2 after hindlimb unloading.** (A) Western blots showing immunoreactivities of MFN2 and ubiquitin. (B) Comparison of immunoreactivity of ubiquitinated MFN2 between 110 and 160 kDa normalized to immunoprecipitated MFN2 levels between groups. CON, control. HU, hindlimb unloading. Data are expressed as means ± SE. *significantly different from CON (p < 0.05).

HU group than that in the CON group (p = 0.049). Therefore, these results indicate that the ubiquitination of MFN2 is decreased in HU-induced soleus muscle atrophy.

## Discussion

In the present study, the expression of nNOS and p-nNOS at Ser1446 was increased in the soleus muscles induced by 14 days of HU. In contrast, the expression of HSP90 and HSP70 was significantly decreased by HU. In addition, our results indicate that HU increased protein oxidative modifications and decreased nitrative modifications. Furthermore, the both S-nitrosylation and nitration of Parkin and the expression of PINK1 and Parkin were unchanged by HU. In contrast, accumulation and reduced ubiquitination of high molecular weight MFN2 were observed in the atrophied soleus muscles. These results may help explain the paradoxical behavior of nNOS and/or NO along with the contribution of the PINK1/Parkin pathway to soleus muscle atrophy.

We showed that protein oxidative modification was increased, whereas protein nitration was decreased by HU (Fig 4). Additionally, we showed that S-nitrosylation was unaltered by HU (Fig 5). These results indicate that peroxynitrite production is decreased and excessive NO production does not occur in atrophied soleus muscles. In contrast, activated nNOS increased in atrophied soleus muscles (Fig 2), indicating that NO production in atrophied soleus muscles does not increase even when activated nNOS is elevated. These findings also suggest that NO does not mediate disuse-induced soleus muscle atrophy after 14 day of HU. The increase in p-nNOS at Ser1446 expression may support a previous study reporting increased nNOS activity in tail suspended mice [15]. In addition, the increased expression of p-nNOS at Ser1446 may have resulted from increased ROS production, given that phosphorylation of nNOS at Ser1446 is induced by ROS [54]. NO production from nNOS, endothelial NOS (eNOS), and inducible NOS (iNOS) decrease with reduced HSP90 concentrations [25,26,55,56]. In addition, NO production in bovines from phosphorylated eNOS at Ser1179, an activated form of eNOS, has been shown to be suppressed by a decrease in HSP90 [57]. The phosphorylation sequence at Ser1446 in nNOS is analogous to that at Ser1179 in bovine eNOS. These findings of previous studies showed that the reduction in NO production due to decreased HSP90 is a common to

all NOS isoforms and that NO production decreases even if NOS is activated. Under our experimental conditions, we observed decreased HSP90 expression in atrophied soleus muscles (Fig 3). Taken together, NO production from p-nNOS at Ser1446 is likely to be suppressed by decreased HSP90 expression in skeletal muscles.

Our results also indicate that increased expression of p-nNOS at Ser1446 accompanied by reduced HSP90 expression might be responsible for disuse-induced soleus muscle atrophy via ROS production from nNOS, namely nNOS uncoupling. In fact, the present results showed that oxidative modifications of proteins were increased (Fig 4). Inhibition of HSP90 in bovine coronary endothelial cells increases superoxide production from activated eNOS [57]. In addition, low levels of HSP90 partly participate in superoxide production in pulmonary artery endothelial cells isolated from 4-week-old lambs [58,59]. These previous findings support the hypothesis that decreased HSP90 expression in the skeletal muscle may also lead to superoxide production from activated nNOS. As already described above, the mediation of disuse-induced skeletal muscle atrophy by nNOS and/or NO is controversial [15,18]. Previous studies showed that knockout and inhibition of nNOS partly prevented disuse-induced skeletal muscle atrophy [15]. This phenomenon may be explained by the suppression of superoxide production from nNOS in atrophied muscle because the inhibition of nNOS reduces superoxide production from nNOS [26]. Therefore, our results regarding the production of ROS by nNOS in disuse-induced soleus muscle atrophy may explain the paradoxical role of nNOS.

In the present study, HU increased the nNOS expression and the p-nNOS at Ser1446 levels in atrophied soleus muscles (Fig 2). These results contradict those of previous studies [15,18]. The difference in the expression of nNOS between the present and previous studies may be related to differences in the biochemical methods used. A previous study showed that the inhibition of HSP90 using specific inhibitors increased the amount of nNOS in the insoluble fraction during protein extraction from cultured cells [27]. This previous finding suggests that nNOS became insoluble at low HSP90 levels [27]. Although whether nNOS becomes insoluble in the atrophied soleus muscles is unknown, we used a lysis buffer containing 8 M urea and 4% CHAPS. This buffer is often used in proteomic analysis to detect many proteins, including membrane proteins, and to solubilize aggregated proteins such as inclusion bodies. A previous study of human soleus muscles after 12 weeks of bed rest using ultrasonication for biochemical analysis has reported increased expression of nNOS [60]. This finding is consistent with our results. Notably, the sonication allows the solubilization of insoluble nNOS during protein extraction [27]. Thus, nNOS may have become insoluble in atrophied soleus muscles, and it is possible that the previous studies could not detect an increase in nNOS expression after 14 days of HU. Such differences in the analysis methods may affect the reproducibility of the experimental results.

A further possibility is that nNOS may accumulate because of attenuated degradation of nNOS in the ubiquitin-proteasome in the atrophied soleus muscles. Ubiquitination and degradation of nNOS are induced by carboxyl terminus of HSP70-interacting protein (CHIP), an E3 ubiquitin ligase [27]. The ubiquitination of nNOS by CHIP is facilitated by the addition of HSP70 [27]. Overexpression of HSP70 in cultured cells promotes ubiquitination of nNOS and results in decreased levels of nNOS protein [41]. In contrast, treatment with an HSP70 inhibitor decreases the binding of HSP70 to nNOS and reduces the ubiquitination of nNOS [41]. We observed that HSP70 expression is decreased in the atrophied soleus muscles in the present study similar to the reported in previous studies (Fig 3) [61,62]. Therefore, the ubiquitination of nNOS might be reduced in the atrophied soleus muscles induced by HU. Meanwhile, nNOS is also degraded by calpain [63], and skeletal muscle disuse activates calpain in atrophied muscle [9,10]. Thus, the contribution of both the ubiquitin-proteasome system and calpain to the degradation of nNOS in atrophied muscles is unknown.

The PINK1/Parkin pathway is affected by both S-nitrosylation and nitration [28,50]. The approximately 40 kDa Parkin detected by IP may be related to the acetone precipitation method used, as the insoluble protein was observed in the IP buffer containing sample proteins after acetone precipitation in both groups. Despite this, the S-nitrosylation and nitration of Parkin were unchanged by HU (Fig 5). These results indicate that Parkin is not inhibited by excessive S-nitrosylation and nitration after 14 days of HU. Additionally, HU did not alter the expression of PINK1 and Parkin (Fig 6). These results suggest that the PINK1/Parkin pathway may not be actively involved in mitophagy in atrophied soleus muscles after 14 days of HU. Previous studies showed that the expression of PINK1 and/or Parkin was not increased in the GAS muscles after muscle inactivity, suggesting that the PINK1/Parkin pathway does not play a major role in skeletal muscle inactivity-induced mitophagy [33–36]. Although the muscles examined in the present study differ from those evaluated in the previous studies [33–36], the PINK1/Parkin pathway may be inactive in atrophied soleus muscles after 14 days of HU.

Analysis of the expression and ubiquitination of MFN2 indicated that soleus muscles with HU-induced atrophy exhibited the high molecular weight MFN2 accumulation and decreased MFN2 ubiquitination (Figs 6 and 7). These findings suggest that the activity of ubiquitin ligases responsible for the ubiquitination of MFN2 is reduced in atrophied soleus muscles. Additionally, these results also support the possibility of the attenuated degradation of some proteins such as nNOS in atrophied muscles. MFN2 is ubiquitinated by various E3 ligases such as Parkin, mitochondrial E3 ubiquitin ligase 1, MARCH5/MITOL, HUWE1, and glycoprotein 78 [64]. We did not determine which pathway is related to the reduced ubiquitination of MFN2. However, a reduction in Parkin activity may contribute to the accumulation and reduced ubiquitination of MFN2 in atrophied soleus muscles. As Parkin is also an HSP70-binding E3 ubiquitin ligase [65], the absence of HSP70 inhibits the Parkin E3 ubiquitin ligase activity [29]. In addition, HSP72 knockout mice showed impaired Parkin activity and mitophagy in the skeletal muscle [66]. Thus, the basal activity of Parkin may be attenuated by reduced HSP70 expression in the atrophied soleus muscles induced by HU.

HSP70 expression is increased in an activity-dependent manner in slow skeletal muscle including the soleus muscles [67], although it is unknown whether HSP90 expression is regulated in an activity-dependent manner. In the present study, the expression of the HSPs was decreased in atrophied soleus muscles induced by HU (Fig 3). This decrease of HSPs in atrophied soleus muscles may result from reduced muscle activity induced by HU. Heat stress used as a treatment for immobilization increases both HSP70 and HSP90 expression in atrophied muscles and prevents skeletal muscle atrophy in rodents and humans [61,68]. Several studies have attempted to elucidate the mechanism to prevent skeletal muscle atrophy by heat stress and/or HSPs [9,69–71]. However, the mechanisms, particularly those of the HSPs-related protein degradation systems discussed above, remain unclear. Thus, further studies are required to investigate the protein degradation systems involving HSPs and the effects of heat stress on these systems in disuse-induced atrophied muscle.

### Limitations of the study

Although the soleus muscle is mainly composed of slow-twitch fibers, a few fast-twitch fibers are also present within in soleus muscles [72]. We did not perform immunohistochemical analysis to determine whether the protein expression observed was specific to slow-twitch fibers. Therefore, the results of this study may not reflect slow-twitch fiber specific molecular events.

We evaluated soleus muscle atrophy after 14 days of HU. A previous study indicated that the expression of nNOS in mice soleus muscles decreased 2 days after HU, but returned to

basal levels at 7 days after HU [73]. Although the expression of HSP90 in atrophied soleus muscles has been reported to decrease after 5 days of HU [74], NO may be involved in promoting skeletal muscle atrophy 7 days after HU. In addition, a previous study has discussed that the expression of HSP70 tends to differ between short-term (5–8 days) and long-term (10 days to 9 weeks) HU [75]. Furthermore, it has also been shown that the molecular mechanism of disuse-induced atrophy may differ at different time points after skeletal muscle disuse [2]. Thus, the molecular events after 14 days of HU observed in the present study may differ from those at other time points after HU.

## Conclusions

The results of the present study indicate that NO production in atrophied soleus muscles does not increase despite activated nNOS levels are elevated after 14 days of HU. These findings suggest that excessive NO production does not mediate disuse-induced soleus muscle atrophy at least after 14 days of HU. Meanwhile, an increase of activated nNOS might be responsible for disuse-induced soleus muscle atrophy through ROS production from nNOS. Moreover, the PINK1/Parkin pathway may not play a role in the mitophagy of atrophied soleus muscles at this time point. In contrast, the activity of Parkin may be downregulated because of reduced HSP70 expression, which may contribute to the accumulation and reduced ubiquitination of a mitochondrial fusion protein in atrophied soleus muscles after 14 days of HU. Therefore, at least partial attenuation of the protein degradation pathway may contribute to soleus muscle atrophy induced by 14 days of HU. The present study provides new insights into the roles of nNOS and a signaling pathway for mitochondrial protein degradation in soleus muscle atrophy.

## Supporting information

**S1 Raw images.**
(PDF)

## Acknowledgments

We would like to thank Editage (www.editage.com) for English language editing.

## Author Contributions

**Conceptualization:** Munehiro Uda.

**Data curation:** Munehiro Uda, Toshinori Yoshihara, Noriko Ichinoseki-Sekine.

**Formal analysis:** Munehiro Uda.

**Funding acquisition:** Munehiro Uda.

**Investigation:** Munehiro Uda, Toshinori Yoshihara, Noriko Ichinoseki-Sekine.

**Methodology:** Munehiro Uda, Toshinori Yoshihara, Noriko Ichinoseki-Sekine.

**Project administration:** Munehiro Uda, Takeshi Baba.

**Resources:** Munehiro Uda, Toshinori Yoshihara, Noriko Ichinoseki-Sekine, Takeshi Baba.

**Supervision:** Munehiro Uda.

**Validation:** Munehiro Uda, Toshinori Yoshihara, Noriko Ichinoseki-Sekine, Takeshi Baba.

**Visualization:** Munehiro Uda.

**Writing – original draft:** Munehiro Uda.

**Writing – review & editing:** Munehiro Uda, Toshinori Yoshihara, Noriko Ichinoseki-Sekine, Takeshi Baba, Toshitada Yoshioka.

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
