## [Decision Letter · Decision Letter 0]

10 Sep 2020

PONE-D-20-25099

Nitric oxide production and a possible mitophagy pathway related to mitochondrial dysfunction in disuse-induced slow skeletal muscle atrophy in adult rats

PLOS ONE

Dear Dr. Uda,

Thank you for submitting your manuscript to PLOS ONE. After careful consideration, we feel that it has merit but does not fully meet PLOS ONE’s publication criteria as it currently stands. Therefore, we invite you to submit a revised version of the manuscript that addresses the points raised during the review process.

We look forward to receiving your revised manuscript.

Kind regards,

Michael Bader

Academic Editor

PLOS ONE

Journal Requirements:

Reviewers' comments:

Reviewer's Responses to Questions

**Comments to the Author**

1. Is the manuscript technically sound, and do the data support the conclusions?

Reviewer #1: No

Reviewer #2: Partly

2. Has the statistical analysis been performed appropriately and rigorously? 

Reviewer #1: Yes

Reviewer #2: Yes

3. Have the authors made all data underlying the findings in their manuscript fully available?

Reviewer #1: Yes

Reviewer #2: Yes

4. Is the manuscript presented in an intelligible fashion and written in standard English?

Reviewer #1: No

Reviewer #2: Yes

5. Review Comments to the Author

Reviewer #1: Uda et al examine the relationship between nitric oxide species and disuse atrophy using the well characterized hindlimb unloading model. They find that hindlimb unloading results in greater nNOS content along with a concurrent lowered content of HSP70 and HSP90. More so, they find differences in different sized MFN2 and tie these differences to altered mitophagy during hindlimb unloading disuse atrophy. The experimental design is appropriate; however, concerns with data reporting as well as overall interpretation of the data limit enthusiasm for the manuscript in current form. Below is a detailed list of suggestions for improvements.

Significant Concerns

1. One of the primary conclusions the authors draw from their data is decreased Pink/Parkin activity is related to mitochondrial dysfunction in disuse atrophy. This argument has a few problems.

First, the authors do not provide data of differences in PINK1 or PARKIN content in the soleus muscle.

Second, the interpretation of MFN2 data based on different isoforms and ubiquitination has limitations. It is unclear if the different bands the authors noted were the result of “true” MFN2 or simply a by-product of non-specific binding, which can occur at a higher rate when using BSA as an initial blocking agent (compared to milk or other commercial products). The company that manufactures this antibody has the molecular weight ~75-85 kDa similar to what other works have described (Su et al Am J Physiol Renal Physiol, 2017; Maclnnis et al J Physiol, 2017; Brown et al J Cachexia Sarcopenia Muscle, 2017). As such it is unclear if these blots at higher kDa are “true” MFN2 isoforms.

Finally, the immunoprecipitation blots of MFN2 appear to not be purely MFN2. If there was immunoprecipitation of MFN2 and blotting for MFN2, this should (at least in my mind) would create one band at ~75-85 kDa; however what the authors present appears more like a traditional ubiquitin blot with bands running the entire column of the gel. I’m not as familiar with IP; but a confirmation blot would be necessary to confirm efficacy of the MFN2 immunoprecipitation.

Because the primary foundation of the mitophagy alterations argument relies on differences in the higher kDa MFN2, the robustness of these conclusions is questionable.

2. The use and rationale of 8 M urea buffer needs to be more clearly explained in the introduction, methods and discussion.

Title:

1. Mitochondrial dysfunction was not specifically measured or quantified in this manuscript, I suggest rewording accordingly.

Abstract:

1. The abstract should have specific data (e.g. 25% less muscle mass in HU v. CON) or something similar.

2. Careful, no oxidative stress was directly measured, markers of oxidative stress is more appropriate language, also which marker specifically?

3. Data does not demonstrate downregulation of Pink1/Parkin

Introduction:

1. “Skeletal muscle atrophy occurs due to both a reduced rate of protein synthesis and an increased rate of degradation” � This sentence should have a reference.

2. Paragraph 2, sentence 3� even if this statement refers to ref 12 I would reference again.

3. End of paragraph 2� In these prior studies was it muscle derived NO? Or from the vascular system, as nitric oxide is also a potent regulator of vasodilation and because the soleus is highly vascularized is it possible that nitric oxide is more of a vascular consequence of HU that is “picked up” in the muscle?

4. “In contrast, phosphorylation of nNOS at Ser847 decreases its activity but produces ROS from nNOS” This sentence is a tad redundant within itself.

5. “Bindings of calmodulin (CaM) and heat shock protein 90 (HSP90) to nNOS is known to promote NO production” Is this good? bad? A quick statement explaining that would be helpful.

6. “Conversely, nNOS produces ROS instead of NO at low levels of HSP90 [23]. Inhibition of HSP90 using specific inhibitors has been shown to increase the amount of nNOS in the insoluble fraction” � insoluble fraction of what? Insoluble fraction is mentioned throughout the manuscript and a few sentences explaining this and why it matters for data interpretation would be helpful.

7. “The ubiquitination of nNOS is also affected by the presence or absence of HSP70” How so?

8. “The signaling pathway is also regulated by NO-related post-translational modification and interaction with a molecular chaperone” Needs a reference

9. “Although the mechanisms of PINK1/Parkin mediated mitophagy have not been fully understood yet, a model elucidating the pathway mechanisms has been shown in a recent review article” �I would use caution here, as this reference appears to be referring to brain research, which probably has some conserved mechanisms between muscle and brain, but may not…

10. “Parkin is also regulated its activity by Snitrosylation, which is the addition of NO to cysteine residues in proteins” Needs a reference

11. “It has already been reported regarding the PINK1/Parkin pathway in several muscles and at different time points after skeletal muscle disuse” � What is the change/effect? Also, probably important to note studies that did not find an effect of disuse on pink1/parkin (Rosa-Caldwell et al APNM, 2020).

12. “However, the participation of that pathway is unclear in the slow skeletal muscle atrophy” “slow” here appears to refer to a time point and not necessarily the muscle phenotype, suggested rewording.

13. “Therefore, together with the changes in the NO production, it is interested to understand whether the PINK1/Parkin pathway participates in disuse-induced slow skeletal muscle atrophy” � Why is it important to focus on predominantly slow twitch fibers? These tend to be more susceptible to disuse compared to other muscle phenotypes, but that should be mentioned in the introduction as rationale for readers who are not as familiar with the disuse literature.

14. “aggregating features of nNOS in atrophied muscle into account” aggregating features is a confusing statement.

Methods:

1. Confirmation of anesthesia would be nice (e.g. no toe pinch reflex).

2. The method of euthanasia should be mentioned.

3. Is this the first time authors have done this protocol in animals? Regardless, I suggest citing prior works that have used similar protocols in the methods to demonstrate external validity.

4. How much protein was loaded for each gel?

5. Tween is generally not recommended for the initial blocking step, is there a particular rationale for its use in this study?

6. The description of antibodies used, company/catalog numbers and dilutions is well documented.

7. A nice bit of additional data could be adding p-Parkin as Pink1 phosphorylates parkin (Nguyen, Padman, & Lazarou, Trends in Cell Biol, 2016).

8. How were membranes stripped? Stripping membranes can inherently create “noise” in the data since most stripping likely removes at least some of the protein.

9. There should be some references in the immunoprecipitation methods.

10. “On the other hand” slightly awkward wording

11. I appreciate the honesty, but going down to 4 may limit statistical power, even if it's post hoc and estimate of power with the new sample size may be nice

Results:

1. The results should have a relative difference between groups (ie 35% lower soleus mass etc).

2. I don't think depicting soleus mass/body mass is necessary as both the denominator and numerator of this ratio change. So I think you can remove it if you would like.

3. Throughout the results the authors use the language “increase” or “decrease”. "Decrease" and "increase" imply longitudinal data, as it appears all data presented are cross sectional, these words should be removed from the results in lieu of "greater" or "lower" or some other words.

4. I'm surprised soleus mass was not normally distributed (that’s the reason for the Mann-Whitney test instead of t-test correct?), maybe to help readers understand how some of these distributions differed, individual data points (especially since there are only two groups) could be insightful.

5. Figure 1 caption� the caption has p<0.01 but in stats section sig was denoted at p<0.05.

6. “Strong immunoreactivity of nNOS was detected in the HU group but was not detected in the CON group” � “Strong” may not be the correct verbiage and I think is more of a discussion level word.

7. I would combine some of these (e.g. Fig 2A and 2C) when referring to nNOS immunoblot data.

8. Some of the figures are a tad fuzzy, don't know if that's due to being changed into a PDF or something but FYI.

9. How proteins were normalized to total proteins should be mentioned earlier in the results section.

10. “These results indicated that nNOS nexpression was increased in atrophied soleus muscles by 14 days of HU, and the increase of nNOS expression was caused by the increased expression of p-nNOS at Ser1446.” � Remove that word “cause” here, this is descriptive data.

11. “Thus, the effects of HU on the expression of CaM” expression isn't really the correct word here, “content” would be more appropriate.

12. “While the immunoreactivity of CaM showed some difference between” � suggested rewording "a mean difference"

13. I believe this is all WB data, perhaps using protein content would be more clear (as opposed to immunoreactivity).

14. “Likewise, the immunoreactivity of 3-NT was weaker in the HU group than in the CON group” These were analyzed by the entire band correct? May be a good idea to say as much.

15. “3-NT content showed a decreasing trend with HU” trend is a specific type of statistical analysis, so this should be reworded.

16. “Thus, these results indicated that ROS production was increased” that's a bit too bold of a statement, as these are all secondary markers and not taken in live tissue this language should be softened.

17. “Although slight differences were observed in several immunoreactive bands of nitrosocysteine” I would remove this, these blots look solidly not different.

18. “These results indicated that excessive NO production does not occur in atrophied soleus muscles and that Parkin was not suppressed by excessive S-nitrosylation and nitration.” Too strong of wording also this is a statement that belongs more in the discussion.

19. Is there a reason to look at MFN2? Was this hypothesized beforehand to be altered? In my opinion, it makes more sense to look at fission related markers, because fission is also needed for effective mitophagy.

20. Also, greater MFN2 content considering the pathology is somewhat counter intuitive since mitophagy/fission is expected to increase and fusion is expected to decrease in these conditions.

21. “MEF2” typo

22. “Therefore, these results indicated that the PINK1/Parkin pathway was downregulated in atrophied soleus muscles induced by HU.” � These may suggest that MFN2 isn't tagged for degradation as much, but it’s unclear what that means.

Discussion:

1. A large portion of the discussion is written with passive language which can limit readability.

2. “NO production from nNOS, endothelial NOS (eNOS), and inducible NOS (iNOS) have been shown to decrease with reduced HSP90 concentrations [22,23,49,50].” some of this is difficult to follow, a few words/phrases to tie these thoughts together would be helpful.

3. “In addition, NO production in bovine from phosphorylated eNOS at

4. Ser1179, an activated form of eNOS, has been shown to be suppressed by a decrease of HSP90” is this inherently bad? I think tying this a bit more with the final sentence would help create a more fluid argument.

5. “In fact, the present results suggest that ROS production is increased, whereas excessive peroxynitrite and NO are not produced in atrophied soleus muscle” � no direct measures of ROS were made in this study, wording should be reflective that these are surrogate markers.

6. “A biochemical property of nNOS is its insolubility under low HSP90 levels” � this should have a reference.

Reviewer #2: In their manuscript, Uda et al. investigated parts of the neuronal nitric oxide synthase (nNOS) and PINK1/Parkin pathway in soleus muscle atrophy induced by 14 days of hindlimb unloading (HU) in adult rats. They performed Western blot analysis of proteins extracted from the soleus muscles and showed that HU increased nNOS expression and phosphorylated nNOS at Ser1446, an activated form of nNOS. HU did not alter nitrosocysteine content and S-nitrosylation of the E3 ligase Parkin, suggesting the absence of excessive nitrosative stress. HU caused an accumulation and reduced the ubiquitination of high molecular weight mitofusin 2 (MFN2), which is a target of Parkin. The authors conclude that despite nNOS activation no excessive NO is produced in atrophied soleus muscles and that the PINK1/Parkin pathway was downregulated under these conditions. The manuscript is well written and addresses an important molecular pathway for muscle atrophy.

However, the data shown only partially support the conclusions drawn. Many of the Western blot experiments shown are low in quality and resolution, and the quantification of the data often do not match the signals seen at the membranes. It is therefore difficult to judge if the conclusions of the manuscript are supported by the data shown. For this reviewer it is also unclear why the 14-day time point was chosen for analyses. Clearly, skeletal muscle atrophy is readily detectable after 14-days of HU. However, signaling pathways that are initiated by HU or any other stress leading to muscle atrophy are activated much earlier. Therefore, an earlier time point would have been more informative for analyses. Uda et al. state that their findings are specific for slow skeletal muscle. In fact, the soleus muscle contains both slow and fast-twitch fibers, and although this muscle is considered to be a slow skeletal muscle in order to state that the investigated pathway is specific for it a comparison to a fast twitch muscle is necessary, or at least a direct comparison of slow and fast fibers in the soleus muscle. Finally, the title of the manuscript mentioning “mitochondrial dysfunction” and the conclusion in the abstract that this work facilitates “the development of effective therapeutic and preventive options against disuse-induced skeletal muscle atrophy” are overstating the findings of the manuscript and must be revised.

Major concerns

1. Although it is tempting to speculate that mitochondrial dysfunction occurs in this model of disuse-induced slow skeletal muscle atrophy this point has not been proven by the authors. Please perform experiments to prove this point.

2. In order to conclude that the nNOS and PINK1/Parkin pathway is specific to HU slow-skeletal muscle atrophy a comparative study needs to be performed that analyses fast-skeletal muscle as well. Please include such data.

3. The time point chosen in the manuscript was 14 days after HU. It is expected that NO production occurs much earlier than after 14 days of HU. Although the 14-day time point is sufficient to detect the muscle atrophy phenotype it is not suitable to decipher the pathways that lead to atrophy. To this reviewer a much earlier time point, hours or few days after HU, would be more appropriate to investigate contributing pathways. If available such data need to be included into the manuscript. Otherwise a clear and referenced reason why the 14-day time point was chosen for analyses must be provided. It would be ideal to provide a time series showing nNOS expression or PINK1/Parkin activity in slow- and fast skeletal muscle during HU. If these data are not available it is not appropriate to conclude that “… NO is not likely to mediate disuse-induced slow skeletal muscle atrophy” as written in the abstract and elsewhere in the manuscript. Because muscle is the primary source of NO in mammals, and muscle loading affects nNOS expression and activation it is expected that changes in muscle loading have immediate effects on muscle nNOS and PINK1/Parkin pathway as well as systemic NO levels. In light of this notion the choice of the 14-day time point is at least questionable.

4. Specific emphasis was put on aggregated proteins and a lysis buffer containing 8 M urea was used to extract proteins from the soleus muscle. The argument that earlier studies have not taken nNOS aggregates into account is not convincing. Please provide a reasoning why this is important and explain why no direct comparison of at least two different loading buffers (i.e. a standard lysis buffer such as RIPA vs. an aggregate solving butter, such as 8 M urea) was performed. It is also uncertain if nNOS aggregates are still functional. Please provided data that support this issue. In addition, although the primary goal was to investigate nNOS aggregates this technique it also affects all the other proteins examined in the manuscript. This could have affected the results obtained and therefore the conclusions drawn and needs further explanation if not additional experimental prove, which I prefer.

5. Since mitochondrial structure and function have not been investigated in this manuscript the conclusion “… downregulation of the PINK1/Parkin pathway may play a role in the mitochondrial dysfunction of these muscles.” is only speculative. If this hypothesis cannot be substantiated by data, it must be deleted.

6. The data showing that skeletal muscle occurred in this model are not very strong. At least histological studies and for the sake of the manuscript, immunohistochemical staining to distinguish fiber-type composition and atrophy must be performed to judge the degree of myofiber atrophy and to state if this was fiber type specific. Likewise, immunohistochemistry could have been performed to investigate the nNOS and PINK1/Parkin pathway and to conclude if any fiber type specific regulation occurred. Please perform such experiments or mention in Limitations.

7. In the manuscript the regulation of the PINK1/Parkin pathway was associated with atrophy. However, that does not mean that this pathway was indeed involved in muscle atrophy. Therefore, the conclusion that PINK1/Parkin pathway is involved in muscle atrophy is not correct. Please rephrase throughout the manuscript.

Minor concerns

1. Page 3, line 3: needs a reference.

2. Page 14, line 9: “MFN2 immunoreactivity between 110 and 160 kDa” This signal might or might not be MFN2. Is there any prove from proteomics data / mass spectrometry that this is indeed MFN2?

3. Page 14, line 4 from bottom: “MEF2” is probably a typo.

4. Please add indicators of molecular weight in all Western blots shown.

5. Figure 7 is not conclusive. It is not sure if any MFN2 signal was detected in A (top panel) as MFN2 is supposed to migrate at 86 kDa. The anti-ubiquitin Blot is not convincing. The decrease in “Ubiquitinated MFN2” (B) appears to result from increased MFN2 160 kDa, which has not even been shown to be MFN2. This data set can be left out of the manuscript.

6. Page 10: “These results indicated that … the increase of nNOS expression was caused by the increased expression of p-nNOS at Ser1446.”. Direct comparison of the Western blot data indicates that there is not a direct correlation between nNOS and p-nNOS Ser1446 so that this conclusion is questionable. It rather appears that the increase in p-nNOS at Ser1446 was caused by an increase in nNOS contents.

6. PLOS authors have the option to publish the peer review history of their article (what does this mean?). If published, this will include your full peer review and any attached files.

Reviewer #1: No

Reviewer #2: No

---

## [Author Response · Author response to Decision Letter 0]

22 Oct 2020

PONE-D-20-25099

Reviewer #1: Uda et al examine the relationship between nitric oxide species and disuse atrophy using the well characterized hindlimb unloading model. They find that hindlimb unloading results in greater nNOS content along with a concurrent lowered content of HSP70 and HSP90. More so, they find differences in different sized MFN2 and tie these differences to altered mitophagy during hindlimb unloading disuse atrophy. The experimental design is appropriate; however, concerns with data reporting as well as overall interpretation of the data limit enthusiasm for the manuscript in current form. Below is a detailed list of suggestions for improvements.

Response: Thank you for the thorough evaluation of our manuscript, and for all of the constructive comments. Please check below, our point-by-point responses.

Significant Concerns

1. One of the primary conclusions the authors draw from their data is decreased Pink/Parkin activity is related to mitochondrial dysfunction in disuse atrophy. This argument has a few problems.

First, the authors do not provide data of differences in PINK1 or PARKIN content in the soleus muscle.

Response: Thank you for your suggestion. We have included the info on PINK1 and Parkin expression in the Abstract of the revised manuscript. In addition, we have deleted the term “mitochondrial dysfunction” from the Title, the last paragraph of Introduction, Discussion, and Conclusion of the revised manuscript. 

Second, the interpretation of MFN2 data based on different isoforms and ubiquitination has limitations. It is unclear if the different bands the authors noted were the result of “true” MFN2 or simply a by-product of non-specific binding, which can occur at a higher rate when using BSA as an initial blocking agent (compared to milk or other commercial products). The company that manufactures this antibody has the molecular weight ~75-85 kDa similar to what other works have described (Su et al Am J Physiol Renal Physiol, 2017; Maclnnis et al J Physiol, 2017; Brown et al J Cachexia Sarcopenia Muscle, 2017). As such it is unclear if these blots at higher kDa are “true” MFN2 isoforms.

Response: Thank you for your comments. In response to the reviewer’s concern, we have rewritten the sentences regarding MFN2 expression and ubiquitination, to improve clarity. 

MFN2 is ubiquitinated at multiple sites (Escobar-Henriques et al., Front. Physiol. (2019) 10:517.). In addition, the binding of ubiquitin moieties to ubiquitinated MFN2 results in the formation of poly-ubiquitin chains. Ubiquitin is an 8.6 kDa protein. When MFN2 is poly-ubiquitinated at multiple sites, its molecular weight is increased. Therefore, high molecular weight MFN2 can be detected. 

In fact, The existence of high molecular weight MFN2 has already been shown in many previous studies in the context of MFN2 ubiquitination and have been used to evaluate MFN2 ubiquitination (Tanaka et al., J Cell Biol. 2010; 191(7): 1367-1380., Ziviani et al. Proc Natl Acad Sci U S A. 2010; 107(11): 5018-5023., Gegg et al., Human Molecular Genetics, 2010, Vol. 19, No. 24 4861–4870., Di Rita et al., Nature Communications (2018)9:3755, DOI: 10.1038/s41467-018-05722-3., Sugiura et al., Molecular Cell 2013; 51(1): 20-34., Chen et al., Am J Physiol Endocrinol Metab 315: E404–E415, 2018.). In addition, a previous study has reported the ubiquitinated MFN2 in skeletal muscles using same antibody used in this study (Kravic et al. Autophagy. 2018;14(2):311-335.). The studies mentioned by the reviewer may have not analyzed MFN2 ubiquitination because these studies did not focus on MFN2 ubiquitination or degradation. Therefore, our study cannot be compared with the studies suggested by the reviewer.

In addition, we examined whether the high molecular weight MFN2 observed in the western blot is the results of non-specific binding by performing immunoprecipitation. In immunoprecipitation, the MFN2 protein present in the whole cell lysate is first pulled down using an antibody. Next, MFN2 is detected using western blotting. We detected high molecular weight MFN2 in this analysis. Thus, we confirmed the immunoreactivity of MFN2 by using different analysis methods. We have briefly explained these points in the revised manuscript as follows:

Expressions of PINK1, Parkin, and MFN2 in the Results section

Page 15, line 14–16.

These results indicated that HU did not alter the expression of PINK1 and Parkin, whereas it may induce the accumulation of high molecular weight MFN2 in the atrophied soleus muscles.

Page 16, line 1–6.

We additionally confirmed whether the high molecular weight MFN2 could be detected by IP. Fig 7A shows the immunoreactivities of MFN2 and ubiquitin after MFN2 IP. Immunoreactivity of MFN2 was observed between 110 and 160 kDa in both groups, which was similar to the results obtained for western blot analysis of MFN2 (Fig 6A). This suggests that the high molecular weight MFN2 is present in the atrophied soleus muscles and proves MFN2 immunoreactivity between 110 and 160 kDa (Fig 6A).

Page 16, line 15–16.

Therefore, these results indicate that the ubiquitination of MFN2 is decreased in HU-induced soleus muscle atrophy.

In the Discussion section

Page 20, line 6–14 

Analysis of the expression and ubiquitination of MFN2 indicated that soleus muscles with HU-induced atrophy exhibited the high molecular weight MFN2 accumulation and decreased MFN2 ubiquitination (Fig 6 and 7). These findings suggest that the activity of ubiquitin ligases responsible for the ubiquitination of MFN2 is reduced in atrophied soleus muscles. Additionally, these results also support the possibility of the attenuated degradation of some proteins such as nNOS in atrophied muscles. MFN2 is ubiquitinated by various E3 ligases such as Parkin, mitochondrial E3 ubiquitin ligase 1, MARCH5/MITOL, HUWE1, and glycoprotein 78 [64]. We did not determine which pathway is related to the reduced ubiquitination of MFN2. However, a reduction in Parkin activity may contribute to the accumulation and reduced ubiquitination of MFN2 in atrophied soleus muscles.

In the Conclusion section

Page 22, line 3–6

In contrast, the activity of Parkin may be downregulated because of reduced HSP70 expression, which may contribute to the accumulation and reduced ubiquitination of a mitochondrial fusion protein in atrophied soleus muscles after 14 days of HU.

Finally, the immunoprecipitation blots of MFN2 appear to not be purely MFN2. If there was immunoprecipitation of MFN2 and blotting for MFN2, this should (at least in my mind) would create one band at ~75-85 kDa; however what the authors present appears more like a traditional ubiquitin blot with bands running the entire column of the gel. I’m not as familiar with IP; but a confirmation blot would be necessary to confirm efficacy of the MFN2 immunoprecipitation.

Because the primary foundation of the mitophagy alterations argument relies on differences in the higher kDa MFN2, the robustness of these conclusions is questionable.

Response: Thank you for your comments. The reviewer’s suggestion requires knockout of the gene encoding for the target protein to confirm the specificity of the antibody. This may be possible in experiments using cultured cells but is difficult in in vivo studies. Although mass spectrometry can be performed, we were unable to use this method because of the lack of samples.

As mentioned above, regarding the band at ~75-85 kDa, MFN2 is ubiquitinated at multiple sites (Escobar-Henriques et al., Front. Physiol. 10:517. (2019)). Additionally, ubiquitin further binds to ubiquitin bound to MFN2 to form poly-ubiquitin chains. The ubiquitin is an 8.6 kDa protein. When MFN2 is poly-ubiquitinated at multiple sites, the molecular weight of MFN2 is increased. Therefore, high molecular weight MFN2 can be detected on ubiquitin blot. In fact, MFN2 has been reported to appear like a ubiquitin ladder in previous studies (Di Rita et al., Nat Commun. 2018, 9: 3755, doi: 10.1038/s41467-018-05722-3., Puri et al., Nat Commun. 2019; 10(1): 3645. doi: 10.1038/s41467-019-11636-5., Sugiura et al., Molecular Cell 2013; 51(1): 20-34., Rouiller et al., Elife. 2018; 7:e32866. doi: 10.7554/eLife.32866., Wauters et al., Autophagy. 2020;16(2): 203-222., Rakovic et al., PLoS One. 2011; 6(3): e16746. doi: 10.1371/journal.pone.0016746.). 

2. The use and rationale of 8 M urea buffer needs to be more clearly explained in the introduction, methods and discussion.

Response: Thank you for your suggestion. We have deleted some sentences related to the 8 M urea buffer from the Introduction, Discussion and Conclusion sections of our revised manuscript. Because our primary goal was not to investigate nNOS aggregates, we have rewritten the Discussion section of our revised manuscript to explain the difference in nNOS expression between the present and previous studies. The rewritten Discussion is as follows:

In the Discussion section

Page 18, line 16–page 19, line 6.

In the present study, HU increased the nNOS expression and the p-nNOS at Ser1446 levels in atrophied soleus muscles (Fig 2). These results contradict those of previous studies [15,18]. The difference in the expression of nNOS between the present and previous studies may be related to differences in the biochemical methods used. A previous study showed that the inhibition of HSP90 using specific inhibitors increased the amount of nNOS in the insoluble fraction during protein extraction from cultured cells [27]. This previous finding suggests that nNOS became insoluble at low HSP90 levels [27]. Although whether nNOS becomes insoluble in the atrophied soleus muscles is unknown, we used a lysis buffer containing 8 M urea and 4％ CHAPS. This buffer is often used in proteomic analysis to detect many proteins, including membrane proteins, and to solubilize aggregated proteins such as inclusion bodies. A previous study of human soleus muscles after 12 weeks of bed rest using ultrasonication for biochemical analysis has reported increased expression of nNOS [60]. This finding is consistent with our results. Notably, the sonication allows the solubilization of insoluble nNOS during protein extraction [27]. Thus, nNOS may have become insoluble in atrophied soleus muscles, and it is possible that the previous studies could not detect an increase in nNOS expression after 14 days of HU. Such differences in the analysis methods may affect the reproducibility of the experimental results.

Title:

1. Mitochondrial dysfunction was not specifically measured or quantified in this manuscript, I suggest rewording accordingly.

Response: Thank you for your suggestion. We reworded the Title of our revised manuscript as follows:

Title: Potential roles of neuronal nitric oxide synthase and the PTEN-induced kinase 1 (PINK1)/Parkin pathway for mitochondrial protein degradation in disuse-induced soleus muscle atrophy in adult rats

Abstract:

1. The abstract should have specific data (e.g. 25% less muscle mass in HU v. CON) or something similar.

Response: Thank you for your suggestion. We have added some specific data. However, we could not include the data specified by the reviewer because of the Abstract word limit.

2. Careful, no oxidative stress was directly measured, markers of oxidative stress is more appropriate language, also which marker specifically?

Response: Thank you for your suggestion. We have updated the Abstract as follows:

Abstract, line 9–11.

Although HU increased malondialdehyde as oxidative modification of the protein, it decreased 6-nitrotryptophan, a marker of protein nitration.

3. Data does not demonstrate downregulation of Pink1/Parkin

Response: Thank you for your suggestion. We have mentioned the expression of PINK1 and Parkin in the Abstract, Results, and Discussion sections of the revised manuscript as follows: 

Abstract, line 12–16.

The expression of PINK1 and Parkin was also unchanged, whereas the expression of heat shock protein 70 (HSP70), which is required for Parkin activity, was reduced in atrophied soleus muscles. Moreover, we observed accumulation and reduced ubiquitination of high molecular weight mitofusin 2, which is a target of Parkin, in atrophied soleus muscles.

Line 18–21

Furthermore, the PINK1/Parkin pathway may not play a role in mitophagy at this time point. In contrast, the activity of Parkin may be downregulated because of reduced HSP70 expression, which may contribute to attenuated degradation of target proteins in the atrophied soleus muscles after 14 days of HU.

Expressions of PINK1, Parkin, and MFN2 in the Results section

Page 15, line 14–15

These results indicated that HU did not alter the expression of PINK1 and Parkin,

First paragraph in the Discussion section

Page 17, line 3–4

Furthermore, the both S-nitrosylation and nitration of Parkin and the expression of PINK1 and Parkin were unchanged by HU.

In the Discussion section

Page 19, line 24–page 20, line 1

Additionally, HU did not alter the expression of PINK1 and Parkin (Fig 6). These results suggest that the PINK1/Parkin pathway may not be actively involved in mitophagy in atrophied soleus muscles after 14 days of HU.

Introduction:

1. “Skeletal muscle atrophy occurs due to both a reduced rate of protein synthesis and an increased rate of degradation” � This sentence should have a reference.

Response: According to the reviewer’s comment, we have added references to support the sentence as follows:

In the Introduction section

Page 3, line 3–4

Skeletal muscle atrophy occurs because of both a reduced rate of protein synthesis and an increased rate of degradation [1-3].

2. Paragraph 2, sentence 3� even if this statement refers to ref 12 I would reference again.

Response: Thank you for your suggestion. According to the reviewer’s comment, we re-referred ref 15 in the sentence as follows:

In the Introduction section

Page 3, line 16–18

This conclusion is based on the observation that both nNOS-null mice and wild type mice receiving an nNOS inhibitor showed a reduced degree of disuse-induced muscle atrophy [15].

3. End of paragraph 2� In these prior studies was it muscle derived NO? Or from the vascular system, as nitric oxide is also a potent regulator of vasodilation and because the soleus is highly vascularized is it possible that nitric oxide is more of a vascular consequence of HU that is “picked up” in the muscle?

Response: Previous studies have reported a decrease in nNOS expression in atrophied soleus muscles, and another study reported NO production in atrophied soleus muscles. However, the effects of eNOS-derived NO were unclear. Additionally, it has been shown that hindlimb unloading induces capillary regression (Roudier et al., J Physiol. 2010; 588: 4579–4591., Kano et al., Acta Physiol Scand. 2000;169(4):271-276.). As we describe in the manuscript, nNOS knockout mice exhibited suppressed muscle atrophy, which we are further examining with a focus on nNOS. 

4. “In contrast, phosphorylation of nNOS at Ser847 decreases its activity but produces ROS from nNOS” This sentence is a tad redundant within itself.

Response: Thank you for your suggestion. We have rewritten the sentences as follows: 

Page 4, line 3–5.

In contrast, phosphorylation of nNOS at Ser847 results in decreases NO production and induced ROS production by nNOS [21,22].

5. “Bindings of calmodulin (CaM) and heat shock protein 90 (HSP90) to nNOS is known to promote NO production” Is this good? bad? A quick statement explaining that would be helpful.

Response: We understand reviewer’s suggestion. However, whether this observation is good or bad depends on the situation. Therefore, it is difficult to expand this point in the revised manuscript.

6. “Conversely, nNOS produces ROS instead of NO at low levels of HSP90 [23]. Inhibition of HSP90 using specific inhibitors has been shown to increase the amount of nNOS in the insoluble fraction” � insoluble fraction of what? Insoluble fraction is mentioned throughout the manuscript and a few sentences explaining this and why it matters for data interpretation would be helpful.

Response: Thank you for your suggestion. We have deleted the sentences from the Introduction, and incorporated them in the Discussion section of the revised manuscript (page 18, line 16–page 19, line 6). 

7. “The ubiquitination of nNOS is also affected by the presence or absence of HSP70” How so?

Response: Thank you for the question. We have rewritten the sentences as follows：

Page 4, line 7–8.

Ubiquitination of nNOS is also reduced in the absence of HSP70 [27].

8. “The signaling pathway is also regulated by NO-related post-translational modification and interaction with a molecular chaperone” Needs a reference

Response: Thank you for your suggestion. We have added references to support in this sentence as follows:

In the Introduction section

Page 4, line 10–11.

The signaling pathway is also regulated by NO-related post-translational modifications and interactions with a molecular chaperone [28,29].

9. “Although the mechanisms of PINK1/Parkin mediated mitophagy have not been fully understood yet, a model elucidating the pathway mechanisms has been shown in a recent review article” �I would use caution here, as this reference appears to be referring to brain research, which probably has some conserved mechanisms between muscle and brain, but may not…

Response: Thank you for your suggestion. We agree with the reviewer. It is unclear whether the same mechanism of PINK1/Parkin mediated mitophagy exist in the muscles and brain. In addition, a review of the PINK1/Parkin pathway in the skeletal muscles has not been published. Therefore, additional studies are required. However, we speculate that the basic mechanism is the same.

10. “Parkin is also regulated its activity by Snitrosylation, which is the addition of NO to cysteine residues in proteins” Needs a reference

Response: Thank you for your suggestion. We have added references to support this sentence s follows:

In the Introduction section

Page 4, line 16–18.

The activity of Parkin is also regulated by S-nitrosylation, which is the addition of NO to cysteine residues in proteins [28].

11. “It has already been reported regarding the PINK1/Parkin pathway in several muscles and at different time points after skeletal muscle disuse” � What is the change/effect? Also, probably important to note studies that did not find an effect of disuse on pink1/parkin (Rosa-Caldwell et al APNM, 2020).

Response: Thank you for your suggestion. We have cited the suggested reference in the revised manuscript and rewritten the Introduction as follows:

Page 4, line 20–page 5, line 3

The involvement of the PINK1/Parkin pathway in skeletal muscle atrophy has been investigated using gastrocnemius (GAS) and tibialis anterior (TA) muscles after skeletal muscle inactivity [32-36]. The expression of PINK1 and/or Parkin in GAS, which is composed of both slow- and fast-twitch fibers, was unchanged or decreased by disuse-induced skeletal muscle atrophy [33-36]. In contrast, their expression in TA, which is composed of predominantly fast-twitch fibers, was increased or remained unchanged after skeletal muscle inactivity [32,36]. However, the role of this pathway in the atrophy of the slow-twitch predominant soleus muscle is unclear. Thus, it is interesting to determine whether the PINK1/Parkin pathway is involved in disuse-induced soleus muscle atrophy together with the potential changes in the production of NO. 

12. “However, the participation of that pathway is unclear in the slow skeletal muscle atrophy” “slow” here appears to refer to a time point and not necessarily the muscle phenotype, suggested rewording.

Response: Thank you for your suggestion. We replaced “slow skeletal muscle” with “soleus muscle” throughout the manuscript. We were also careful when we mentioned muscle phenotypes, for the sake of readability.

13. “Therefore, together with the changes in the NO production, it is interested to understand whether the PINK1/Parkin pathway participates in disuse-induced slow skeletal muscle atrophy” � Why is it important to focus on predominantly slow twitch fibers? These tend to be more susceptible to disuse compared to other muscle phenotypes, but that should be mentioned in the introduction as rationale for readers who are not as familiar with the disuse literature.

Response: Thank you for your suggestion. We have rewritten the Introduction as follows: 

Page 4, line 20–page 5, line 3

The involvement of the PINK1/Parkin pathway in skeletal muscle atrophy has been investigated using gastrocnemius (GAS) and tibialis anterior (TA) muscles after skeletal muscle inactivity [32-36]. The expression of PINK1 and/or Parkin in GAS, which is composed of both slow- and fast-twitch fibers, was unchanged or decreased by disuse-induced skeletal muscle atrophy [33-36]. In contrast, their expression in TA, which is composed of predominantly fast-twitch fibers, was increased or remained unchanged after skeletal muscle inactivity [32,36]. However, the role of this pathway in the atrophy of the slow-twitch predominant soleus muscle is unclear. Thus, it is interesting to determine whether the PINK1/Parkin pathway is involved in disuse-induced soleus muscle atrophy together with the potential changes in the production of NO. 

14. “aggregating features of nNOS in atrophied muscle into account” aggregating features is a confusing statement.

Response: Thank you for your suggestion. We have deleted the sentences from the Introduction section of our revised manuscript because this statement was confusing.

Methods:

1. Confirmation of anesthesia would be nice (e.g. no toe pinch reflex).

Response: Thank you for your suggestion. We have explained this point as follows:

Page 6, line 11–13

Rats from the CON and HU groups were deeply anesthetized with pentobarbital sodium. Once the rats became completely unresponsive to stimulation, the soleus muscles were removed and frozen in liquid nitrogen.

2. The method of euthanasia should be mentioned.

Response: Thank you for your suggestion. We have added the method of euthanasia as follows:

Page 6, line 13

Then, the rats were euthanized by exsanguination.

3. Is this the first time authors have done this protocol in animals? Regardless, I suggest citing prior works that have used similar protocols in the methods to demonstrate external validity.

Response: Thank you for your suggestion. We have cited our previous work.

Page 6, line 13–14

Protein extraction and biochemical analysis were performed in a manner similar to that reported in our previous studies [37,38].

4. How much protein was loaded for each gel?

Response: Thank you for the question. We have disclosed this information in the revised manuscript as follows:

Page 6, line 21–22

Equal amounts of proteins (30-200 μg/lane) were loaded onto a polyacrylamide gels, and the proteins were separated using sodium dodecyl sulfate gel electrophoresis (SDS-PAGE).

5. Tween is generally not recommended for the initial blocking step, is there a particular rationale for its use in this study?

Response: We and the reviewer appear to have different perceptions of blocking agents. Tween-20 is often used in blocking solutions and has been widely used in many previous studies. Therefore, it is standard practice to use Tween-20 in these applications. The particular reason for the use of Tween 20 was that the blocking solution containing this reagent led to clearer results. 

6. The description of antibodies used, company/catalog numbers and dilutions is well documented.

Response: Thank you very much. We did our best to ensure the reproducibility of our study by others if they decide to do so.

7. A nice bit of additional data could be adding p-Parkin as Pink1 phosphorylates parkin (Nguyen, Padman, & Lazarou, Trends in Cell Biol, 2016).

Response: Thank you for your suggestion. PINK1 activates Parkin by phosphorylating it at the Ser65 residue. Thus, an anti-phospholyrated Ser65 Parkin antibody can enable us to analyze the activation of Parkin. However, anti-phosphorylated Ser65 Parkin antibodies are currently commercially unavailable from commercial sources.

8. How were membranes stripped? Stripping membranes can inherently create “noise” in the data since most stripping likely removes at least some of the protein.

Response: We used a commercially available stripping buffer (WB Stripping Solution Strong, 05677-65, Nacalai Tesque). As the reviewer pointed out, it is possible that stripping could have resulted in protein loss. However, WB can detect the reaction even after several stripping steps.

9. There should be some references in the immunoprecipitation methods.

Response: Thank you for your suggestion. We performed immunoprecipitation according to the method described by BIO-RAD. We have cited our previous work involving immunoprecipitation as follows:

Page 6, line 13–14 

Protein extraction and biochemical analysis were performed in a manner similar to that reported in our previous studies [37,38].

10. “On the other hand” slightly awkward wording

Response: Thank you for your suggestion. we have changed the term from “on the other hand” to “in contrast”. (page 8, line 13)

11. I appreciate the honesty, but going down to 4 may limit statistical power, even if it's post hoc and estimate of power with the new sample size may be nice

Response: We have shown all data according to the PLOS one policy. We agree with the reviewer’s suggestion regarding statistical power. 

Results:

1. The results should have a relative difference between groups (ie 35% lower soleus mass etc).

Response: Thank you for your suggestion. We have included the relative differences between groups in the results section of the revised manuscript.

2. I don't think depicting soleus mass/body mass is necessary as both the denominator and numerator of this ratio change. So I think you can remove it if you would like.

　

Response: Thank you for your suggestion. The evaluation of soleus mass/body mass is necessary to eliminate the effects of body size, such as that when introduced upon including the large rats into a CON group and small rats in the HU group. In such a case, body size affects muscle weight because large rats have large muscles, which may lead to overestimate of the degree of muscle atrophy.

3. Throughout the results the authors use the language “increase” or “decrease”. "Decrease" and "increase" imply longitudinal data, as it appears all data presented are cross sectional, these words should be removed from the results in lieu of "greater" or "lower" or some other words.

Response: Thank you for your suggestion. We changed these terms as much as possible.

4. I'm surprised soleus mass was not normally distributed (that’s the reason for the Mann-Whitney test instead of t-test correct?), maybe to help readers understand how some of these distributions differed, individual data points (especially since there are only two groups) could be insightful.

Response: Thank you for your suggestion. In the HU group, the mean soleus muscle weight was 54.6 mg. Individual soleus muscle weights were 56.3, 56.5, 52.9, 52.9, 52.4, and 56.3mg. Drawing a normal distribution graph using these data results in two peaks. Therefore, SPSS analysis did not indicate normal distribution.

5. Figure 1 caption� the caption has p<0.01 but in stats section sig was denoted at p<0.05.

Response: We set statistical significance at p < 0.05, meaning all values below 0.05 are significant. Thus, p < 0.01 indicates statistical significance.

6. “Strong immunoreactivity of nNOS was detected in the HU group but was not detected in the CON group” � “Strong” may not be the correct verbiage and I think is more of a discussion level word.

Response: Thank you for your suggestion. We would like to show the observed density of the band.

7. I would combine some of these (e.g. Fig 2A and 2C) when referring to nNOS immunoblot data.

Response: Thank you for your suggestion. This study does not contain large amounts of data. Therefore, it does not seem necessary to combine the figures.

8. Some of the figures are a tad fuzzy, don't know if that's due to being changed into a PDF or something but FYI.

Response: We consider that this may have occurred during the conversion to a PDF file because the image is not blurred when viewed on a PC screen, even when it is enlarged. The submitted figures align with the journal quality standards.

9. How proteins were normalized to total proteins should be mentioned earlier in the results section.

Response: Thank you for your suggestion. We normalize the protein expression levels to the total protein. However, some proteins were normalized against total and non-phosphorylated proteins. Moreover, proteins in which multiple bands were observed, were normalized to those with several bands added together. Thus, we have explained this in each instance to avoid confusion to the reader.

10“These results indicated that nNOS nexpression was increased in atrophied soleus muscles by 14 days of HU, and the increase of nNOS expression was caused by the increased expression of p-nNOS at Ser1446.” � Remove that word “cause” here, this is descriptive data.

Response: Thank you for your suggestion. We have rewritten the sentences as follows: 

Page 11, line 2–3

These results indicate nNOS expression and p-nNOS at Ser1446 were increased in response to 14 days of HU-induced atrophy in soleus muscles.

11. “Thus, the effects of HU on the expression of CaM” expression isn't really the correct word here, “content” would be more appropriate.

Response: Thank you for your suggestion. We have rewritten the term as follows:

Page 11, line 16–17.

Thus, the effects of HU on the content of CaM, HSP90, and HSP70 were also evaluated in this study.

12. “While the immunoreactivity of CaM showed some difference between” � suggested rewording "a mean difference"

Response: Thank you for your suggestion. We have rewritten the sentence as follows:

Page 11, line 17–18

The immunoreactivity of CaM appeared to be slightly different between the CON and HU groups (Fig 3A),

13. I believe this is all WB data, perhaps using protein content would be more clear (as opposed to immunoreactivity).

Response: As the reviewer suggested, we have shown all WB data in accordance with the PLOS ONE policy. The protein expression levels were normalized to total protein levels. Additionally, p-nNOS and MFN2 were normalized to unphosphorylated nNOS and total MFN2 expression levels, respectively.

14. “Likewise, the immunoreactivity of 3-NT was weaker in the HU group than in the CON group” These were analyzed by the entire band correct? May be a good idea to say as much.

Response: Thank you for your question. We analyzed these data from the top to the bottom of the lane.

15. “3-NT content showed a decreasing trend with HU” trend is a specific type of statistical analysis, so this should be reworded.

Response: Thank you for your suggestion. We have rewritten the sentence as follows:

Page 12, line 24

3-NT content tended to be lower with HU (p = 0.176, Fig 4G).

16. “Thus, these results indicated that ROS production was increased” that's a bit too bold of a statement, as these are all secondary markers and not taken in live tissue this language should be softened.

Response: Thank you for your suggestion. We have updated the Results and Discussion section of our revised manuscript as follows:

Page 12, line 24–page 13, line 1

Thus, these results indicate that while HU increased oxidative modification of the protein, it decreased protein nitration in atrophied soleus muscles.

First paragraph of the Discussion section

Page 17, line 2–3

In addition, our results indicate that HU increased protein oxidative modifications and decreased nitrative modifications.

In the Discussion section

Page 17, line 8–11

We showed that protein oxidative modification was increased, whereas protein nitration was decreased by HU (Fig 4). Additionally, we showed that S-nitrosylation was unaltered by HU (Fig 5). These results indicate that peroxynitrite production is decreased and excessive NO production does not occur in atrophied soleus muscles.

17. “Although slight differences were observed in several immunoreactive bands of nitrosocysteine” I would remove this, these blots look solidly not different.

Response: Thank you for your suggestion. We have deleted the sentence.

18. “These results indicated that excessive NO production does not occur in atrophied soleus muscles and that Parkin was not suppressed by excessive S-nitrosylation and nitration.” Too strong of wording also this is a statement that belongs more in the discussion.

Response: Thank you for your suggestion. We have updated the Results and Discussion sections of our revised manuscript as follows:

Page 13, line 25–page 14, line 1 

These results indicate that the S-nitrosylation and nitration of Parkin were unchanged by HU.

First paragraph in the Discussion section

Page 17, line 3–4

Furthermore, the both S-nitrosylation and nitration of Parkin and the expression of PINK1 and Parkin were unchanged by HU.

In the Discussion section

Page 19, line 22–24

Despite this, the S-nitrosylation and nitration of Parkin were unchanged by HU (Fig 5). These results indicate that Parkin is not inhibited by excessive S-nitrosylation and nitration after 14 days of HU.

19. Is there a reason to look at MFN2? Was this hypothesized beforehand to be altered? In my opinion, it makes more sense to look at fission related markers, because fission is also needed for effective mitophagy.

Response: Thank you for your suggestion. MFN2 is a target of Parkin. Therefore, we analyzed MFN2. However, we agree with the reviewer’s suggestion that proteins associated with mitochondrial fission can also serve as mitophagy markers. 

20. Also, greater MFN2 content considering the pathology is somewhat counter intuitive since mitophagy/fission is expected to increase and fusion is expected to decrease in these conditions.

Response: Thank you for your suggestion. Mitophagy is a complex event that is regulated by multiple protein interactions. It is unknown how up- and downregulation of MFN2 affect mitophagy. A recent study showed that the upregulation of MFN2 expression is associated to mitochondrial fragmentation in neurons (Puri et al., Nat Commun. 2019; 10: 3645.). Thus, the result may not be inconsistent. Additional studies of skeletal muscle cells are required.

21. “MEF2” typo

Response: Thank you for pointing this out. This error has been corrected (page 15, line 9).

22. “Therefore, these results indicated that the PINK1/Parkin pathway was downregulated in atrophied soleus muscles induced by HU.” � These may suggest that MFN2 isn't tagged for degradation as much, but it’s unclear what that means.

Response: Thank you for your suggestion. We have rewritten the sentence as follows:

Page 16, line 15–16

Therefore, these results indicate that the ubiquitination of MFN2 is decreased in HU-induced soleus muscle atrophy.

Discussion:

1. A large portion of the discussion is written with passive language which can limit readability.

Response: Thank you for your suggestion. We have rewritten some of the passive sentences.

2. “NO production from nNOS, endothelial NOS (eNOS), and inducible NOS (iNOS) have been shown to decrease with reduced HSP90 concentrations [22,23,49,50].” some of this is difficult to follow, a few words/phrases to tie these thoughts together would be helpful.

Response: Thank you for your suggestion. We have deleted the sentence.

3. “In addition, NO production in bovine from phosphorylated eNOS at Ser1179, an activated form of eNOS, has been shown to be suppressed by a decrease of HSP90” is this inherently bad? I think tying this a bit more with the final sentence would help create a more fluid argument.

Response: Thank you for your suggestion. We have added a sentence that relates the sentences to the points mentioned above and the final sentence as follows:

Page 17, line 21–23 

These findings of previous studies showed that the reduction in NO production due to decreased HSP90 is a common to all NOS isoforms and that NO production decreases even if NOS is activated.

4. “In fact, the present results suggest that ROS production is increased, whereas excessive peroxynitrite and NO are not produced in atrophied soleus muscle” � no direct measures of ROS were made in this study, wording should be reflective that these are surrogate markers.

Response: Thank you for your suggestion. We have rewritten the sentence as follows: 

Page 18, line 4–5

In fact, the present results showed that oxidative modifications of proteins were increased (Fig 4).

5. “A biochemical property of nNOS is its insolubility under low HSP90 levels” � this should have a reference.

Response: Thank you for your suggestion. We have deleted the sentence and rewritten the Discussion of our revised manuscript as follows: 

Page 18, line 16–page 19, line 6 

In the present study, HU increased the nNOS expression and the p-nNOS at Ser1446 levels in atrophied soleus muscles (Fig 2). These results contradict those of previous studies [15,18]. The difference in the expression of nNOS between the present and previous studies may be related to differences in the biochemical methods used. A previous study showed that the inhibition of HSP90 using specific inhibitors increased the amount of nNOS in the insoluble fraction during protein extraction from cultured cells [27]. This previous finding suggests that nNOS became insoluble at low HSP90 levels [27]. Although whether nNOS becomes insoluble in the atrophied soleus muscles is unknown, we used a lysis buffer containing 8 M urea and 4％ CHAPS. This buffer is often used in proteomic analysis to detect many proteins, including membrane proteins, and to solubilize aggregated proteins such as inclusion bodies. A previous study of human soleus muscles after 12 weeks of bed rest using ultrasonication for biochemical analysis has reported increased expression of nNOS [60]. This finding is consistent with our results. Notably, the sonication allows the solubilization of insoluble nNOS during protein extraction [27]. Thus, nNOS may have become insoluble in atrophied soleus muscles, and it is possible that the previous studies could not detect an increase in nNOS expression after 14 days of HU. Such differences in the analysis methods may affect the reproducibility of the experimental results.

Reviewer #2: In their manuscript, Uda et al. investigated parts of the neuronal nitric oxide synthase (nNOS) and PINK1/Parkin pathway in soleus muscle atrophy induced by 14 days of hindlimb unloading (HU) in adult rats. They performed Western blot analysis of proteins extracted from the soleus muscles and showed that HU increased nNOS expression and phosphorylated nNOS at Ser1446, an activated form of nNOS. HU did not alter nitrosocysteine content and S-nitrosylation of the E3 ligase Parkin, suggesting the absence of excessive nitrosative stress. HU caused an accumulation and reduced the ubiquitination of high molecular weight mitofusin 2 (MFN2), which is a target of Parkin. The authors conclude that despite nNOS activation no excessive NO is produced in atrophied soleus muscles and that the PINK1/Parkin pathway was downregulated under these conditions. The manuscript is well written and addresses an important molecular pathway for muscle atrophy.

Response: Thank you for the thorough evaluation of our manuscript, and for all of the constructive comments. Please check below, our point-by-point responses.

However, the data shown only partially support the conclusions drawn. Many of the Western blot experiments shown are low in quality and resolution, and the quantification of the data often do not match the signals seen at the membranes. It is therefore difficult to judge if the conclusions of the manuscript are supported by the data shown. For this reviewer it is also unclear why the 14-day time point was chosen for analyses. Clearly, skeletal muscle atrophy is readily detectable after 14-days of HU. However, signaling pathways that are initiated by HU or any other stress leading to muscle atrophy are activated much earlier. Therefore, an earlier time point would have been more informative for analyses. Uda et al. state that their findings are specific for slow skeletal muscle. In fact, the soleus muscle contains both slow and fast-twitch fibers, and although this muscle is considered to be a slow skeletal muscle in order to state that the investigated pathway is specific for it a comparison to a fast twitch muscle is necessary, or at least a direct comparison of slow and fast fibers in the soleus muscle. Finally, the title of the manuscript mentioning “mitochondrial dysfunction” and the conclusion in the abstract that this work facilitates “the development of effective therapeutic and preventive options against disuse-induced skeletal muscle atrophy” are overstating the findings of the manuscript and must be revised.

Major concerns

1. Although it is tempting to speculate that mitochondrial dysfunction occurs in this model of disuse-induced slow skeletal muscle atrophy this point has not been proven by the authors. Please perform experiments to prove this point.

Response: Thank you for your suggestion. As the reviewer stated, we did not analyze mitochondrial dysfunction. Therefore, according to the reviewer’s comment, we have deleted the term "mitochondrial dysfunction" from the Title, the conclusion in Abstract, the last paragraph of Introduction, Discussion, and Conclusion of the revised manuscript. Additionally, we have revised these sections.

2. In order to conclude that the nNOS and PINK1/Parkin pathway is specific to HU slow-skeletal muscle atrophy a comparative study needs to be performed that analyses fast-skeletal muscle as well. Please include such data.

Response: Thank you for the helpful comments on our paper. Because of our lack of writing skills, our intended point was not clear. We did not intend to claim that nNOS and PINK1/Parkin were involved in the atrophy of only slow-twitch fibers. Therefore, we added a limitation of the study to the Discussion to address your concerns regarding the fiber type composition. In addition, we have changed the phrase from “slow skeletal muscle atrophy “ to “soleus muscle atrophy”. Previous studies have investigated the PINK1/Parkin pathway in fast-twitch fiber predominant skeletal muscles such as tibialis anterior muscle. In contrast, we used soleus, which is a slow-twitch fiber predominant skeletal muscle. Thus, we have used the term “slow skeletal muscle atrophy” The following information has been added to the Discussion.

Limitation in the Discussion section

Page 21, line 7–10

Although the soleus muscle is mainly composed of slow-twitch fibers, a few fast-twitch fibers are also present within in soleus muscles [72]. We did not perform immunohistochemical analysis to determine whether the protein expression observed was specific to slow-twitch fibers. Therefore, the results of this study may not reflect slow-twitch fiber specific molecular events.

3. The time point chosen in the manuscript was 14 days after HU. It is expected that NO production occurs much earlier than after 14 days of HU. Although the 14-day time point is sufficient to detect the muscle atrophy phenotype it is not suitable to decipher the pathways that lead to atrophy. To this reviewer a much earlier time point, hours or few days after HU, would be more appropriate to investigate contributing pathways. If available such data need to be included into the manuscript. Otherwise a clear and referenced reason why the 14-day time point was chosen for analyses must be provided. It would be ideal to provide a time series showing nNOS expression or PINK1/Parkin activity in slow- and fast skeletal muscle during HU. If these data are not available it is not appropriate to conclude that “… NO is not likely to mediate disuse-induced slow skeletal muscle atrophy” as written in the abstract and elsewhere in the manuscript. Because muscle is the primary source of NO in mammals, and muscle loading affects nNOS expression and activation it is expected that changes in muscle loading have immediate effects on muscle nNOS and PINK1/Parkin pathway as well as systemic NO levels. In light of this notion the choice of the 14-day time point is at least questionable.

Response: Thank you for your suggestion. The studies referenced used 14-days tail suspension and excessive production of NO promotes muscle atrophy (Suzuki et al., J Clin Invest. 2007; 117(9): 2468-2476.) or that NO was not produced in atrophied soleus muscles (Lomonosova et al., Biochemistry (Mosc). 2011; 76(5): 571-80.) at that time point. Therefore, we used the HU for 14 days. In this respect, we added sentences to the Method section to describe why this time point was chosen. Additionally, we have added content in the Introduction section regarding the time-points used for muscle harvest in previous studies. 

However, we agree with the reviewer’s point that NO may promote skeletal muscle atrophy in the early phase in addition to 14 days of muscle inactivity. A previous study indicated that nNOS expression in mice soleus muscles transiently decreased at 2 days after and returned to basal levels at 7 days after unloading (Lechado et al., J Pathol. 2018; 246(4): 433-446.). Therefore, NO may promote skeletal muscle atrophy as early as 7 days after hindlimb unloading. Thus, we revised the sentences indicated by the reviewer to limit their claim. Additionally, we added the time-point as a limitation of the present study to the discussion section of revised manuscript. The revisions in the manuscript are as follows: 

In the Introduction section

Page 3, line 14–16.

A previous study has indicated that nNOS and/or NO mediate 14 days of tail suspension-induced skeletal muscle atrophy by activating of muscle-specific RING finger protein 1 and muscle atrophy F-box 1/atorogen-1, which are E3 ubiquitin ligases [15].

Page 3, line 21–23.

In contrast, other previous studies showed that the expression of nNOS and the production of NO is decreased in atrophied soleus muscles after 10 to 14 days of HU [18,19].

Hindlimb unloading in the Methods section 

Page 6, line 3–4 

We chose this time point because, in previous studies, NO production and nNOS expression were evaluated in isolated muscles after 14 days of HU [15,18].

Limitations of the study in the Discussion section 

Page 21, line 11–19

We evaluated soleus muscle atrophy after 14 days of HU. A previous study indicated that the expression of nNOS in mice soleus muscles decreased 2 days after HU, but returned to basal levels at 7 days after HU [73]. Although the expression of HSP90 in atrophied soleus muscles has been reported to decrease after 5 days of HU [74], NO may be involved in promoting skeletal muscle atrophy 7 days after HU. In addition, a previous study has discussed that the expression of HSP70 tends to differ between short-term (5-8 days) and long-term (10 days to 9 weeks) HU [75]. Furthermore, it has also been shown that the molecular mechanism of disuse-induced atrophy may differ at different time points after skeletal muscle disuse [2]. Thus, the molecular events after 14 days of HU observed in the present study may differ from those at other time points after HU.

4. Specific emphasis was put on aggregated proteins and a lysis buffer containing 8 M urea was used to extract proteins from the soleus muscle. The argument that earlier studies have not taken nNOS aggregates into account is not convincing. Please provide a reasoning why this is important and explain why no direct comparison of at least two different loading buffers (i.e. a standard lysis buffer such as RIPA vs. an aggregate solving butter, such as 8 M urea) was performed. It is also uncertain if nNOS aggregates are still functional. Please provided data that support this issue. In addition, although the primary goal was to investigate nNOS aggregates this technique it also affects all the other proteins examined in the manuscript. This could have affected the results obtained and therefore the conclusions drawn and needs further explanation if not additional experimental prove, which I prefer.

Response: Thank you for your suggestion. According to the reviewer’s comment, we deleted the sentences describing the urea buffer from the Introduction section. In addition, we deleted the sentence “Furthermore, we predict that the elevation of nNOS in atrophied soleus muscle may be due to an HU-induced increase in poorly soluble and possibly aggregated nNOS.” from the Conclusion section of the revised manuscript.

This misrepresented our intention because of our lack of writing skills. The primary goal was not to investigate nNOS aggregates. Therefore, we did not compare experiments using different lysis buffers. However, we are planning the experiments aimed at evaluating insoluble nNOS and the carboxyl terminus of HSP70-interacting protein (CHIP), and we thus would like to report the experiments in a future paper. 

Because primary goal was not to investigate nNOS aggregates, we have rewritten the Discussion section of our revised manuscript to explain the difference in nNOS expression between the present and previous studies. Additionally, we have deleted the sentence explaining nNOS insolubility and aggregation from the Introduction section and rewrote the Discussion as follows:

In the Discussion section

Page 18, line 16–page 19, line 6

In the present study, HU increased the nNOS expression and the p-nNOS at Ser1446 levels in atrophied soleus muscles (Fig 2). These results contradict those of previous studies [15,18]. The difference in the expression of nNOS between the present and previous studies may be related to differences in the biochemical methods used. A previous study showed that the inhibition of HSP90 using specific inhibitors increased the amount of nNOS in the insoluble fraction during protein extraction from cultured cells [27]. This previous finding suggests that nNOS became insoluble at low HSP90 levels [27]. Although whether nNOS becomes insoluble in the atrophied soleus muscles is unknown, we used a lysis buffer containing 8 M urea and 4％ CHAPS. This buffer is often used in proteomic analysis to detect many proteins, including membrane proteins, and to solubilize aggregated proteins such as inclusion bodies. A previous study of human soleus muscles after 12 weeks of bed rest using ultrasonication for biochemical analysis has reported increased expression of nNOS [60]. This finding is consistent with our results. Notably, the sonication allows the solubilization of insoluble nNOS during protein extraction [27]. Thus, nNOS may have become insoluble in atrophied soleus muscles, and it is possible that the previous studies could not detect an increase in nNOS expression after 14 days of HU. Such differences in the analysis methods may affect the reproducibility of the experimental results.

5. Since mitochondrial structure and function have not been investigated in this manuscript the conclusion “… downregulation of the PINK1/Parkin pathway may play a role in the mitochondrial dysfunction of these muscles.” is only speculative. If this hypothesis cannot be substantiated by data, it must be deleted.

Response: Thank you for your suggestion. According to the reviewer’s comment, we have deleted the sentence “the downregulation of the PINK1/Parkin pathway may play a role in the mitochondrial dysfunction of these muscles.” from the abstract of our revised manuscript. In addition, the related text has been removed from the Discussion and Conclusions of the revised manuscript. We have also rewritten the Title, Abstract and Conclusion of the revised manuscript as follows: 

Title

Potential roles of neuronal nitric oxide synthase and the PTEN-induced kinase 1 (PINK1)/Parkin pathway for mitochondrial protein degradation in disuse-induced soleus muscle atrophy in adult rats

Abstract

Excessive nitric oxide (NO) production and mitochondrial dysfunction can activate protein degradation in disuse-induced skeletal muscle atrophy. However, the increase in NO production in atrophied muscles remains controversial. In addition, although several studies have investigated the PTEN-induced kinase 1 (PINK1)/Parkin pathway, a mitophagy pathway, in atrophied muscle, the involvement of this pathway in soleus muscle atrophy is unclear. In this study, we investigated the involvement of neuronal nitric oxide synthase (nNOS) and the PINK1/Parkin pathway in soleus muscle atrophy induced by 14 days of hindlimb unloading (HU) in adult rats. HU lowered the weight of the soleus muscles. nNOS expression showed an increase in atrophied soleus muscles. Although HU increased malondialdehyde as oxidative modification of the protein, it decreased 6-nitrotryptophan, a marker of protein nitration. Additionally, the nitrosocysteine content and S-nitrosylated Parkin were not altered, suggesting the absence of excessive nitrosative stress after HU. The expression of PINK1 and Parkin was also unchanged, whereas the expression of heat shock protein 70 (HSP70), which is required for Parkin activity, was reduced in atrophied soleus muscles. Moreover, we observed accumulation and reduced ubiquitination of high molecular weight mitofusin 2, which is a target of Parkin, in atrophied soleus muscles. These results indicate that excessive NO is not produced in atrophied soleus muscles despite nNOS accumulation, suggesting that excessive NO dose not mediate in soleus muscle atrophy at least after 14 days of HU. Furthermore, the PINK1/Parkin pathway may not play a role in mitophagy at this time point. In contrast, the activity of Parkin may be downregulated because of reduced HSP70 expression, which may contribute to attenuated degradation of target proteins in the atrophied soleus muscles after 14 days of HU. The present study provides new insights into the roles of nNOS and a protein degradation pathway in soleus muscle atrophy.

Conclusion

The results of the present study indicate that NO production in atrophied soleus muscles does not increase despite activated nNOS levels are elevated after 14 days of HU. These findings suggest that excessive NO production does not mediate disuse-induced soleus muscle atrophy at least after 14 days of HU. Meanwhile, an increase of activated nNOS might be responsible for disuse-induced soleus muscle atrophy through ROS production from nNOS. Moreover, the PINK1/Parkin pathway may not play a role in the mitophagy of atrophied soleus muscles at this time point. In contrast, the activity of Parkin may be downregulated because of reduced HSP70 expression, which may contribute to the accumulation and reduced ubiquitination of a mitochondrial fusion protein in atrophied soleus muscles after 14 days of HU. Therefore, at least partial attenuation of the protein degradation pathway may contribute to soleus muscle atrophy induced by 14 days of HU. The present study provides new insights into the roles of nNOS and a signaling pathway for mitochondrial protein degradation in soleus muscle atrophy.

6. The data showing that skeletal muscle occurred in this model are not very strong. At least histological studies and for the sake of the manuscript, immunohistochemical staining to distinguish fiber-type composition and atrophy must be performed to judge the degree of myofiber atrophy and to state if this was fiber type specific. Likewise, immunohistochemistry could have been performed to investigate the nNOS and PINK1/Parkin pathway and to conclude if any fiber type specific regulation occurred. Please perform such experiments or mention in Limitations.

Response: We have discussed these concerns about the fiber type composition as follows:

Limitations of the study in Discussion section

Page 21, line 7–10

Although the soleus muscle is mainly composed of slow-twitch fibers, a few fast-twitch fibers are also present within in soleus muscles [72]. We did not perform immunohistochemical analysis to determine whether the protein expression observed was specific to slow-twitch fibers. Therefore, the results of this study may not reflect slow-twitch fiber specific molecular events.

7. In the manuscript the regulation of the PINK1/Parkin pathway was associated with atrophy. However, that does not mean that this pathway was indeed involved in muscle atrophy. Therefore, the conclusion that PINK1/Parkin pathway is involved in muscle atrophy is not correct. Please rephrase throughout the manuscript.

Response: Thank you for your suggestion. we have revised the description of the involvement of the PINK1/Parkin pathway in soleus muscle atrophy. Additionally, we have revised the text related to the PINK1/Parkin pathway as follows: 

In the Abstract section

Line 18–21.

Furthermore, the PINK1/Parkin pathway may not play a role in mitophagy at this time point. In contrast, the activity of Parkin may be downregulated because of reduced HSP70 expression, which may contribute to attenuated degradation of target proteins in the atrophied soleus muscles after 14 days of HU.

Expressions of PINK1 Parkin, and MFN2 in the Results section

Page 15, line 14–16

These results indicated that HU did not alter the expression of PINK1 and Parkin, whereas it may induce the accumulation of high molecular weight MFN2 in the atrophied soleus muscles. 

First paragraph in the Discussion section 

Page 17, line 3–4

Furthermore, the both S-nitrosylation and nitration of Parkin and the expression of PINK1 and Parkin were unchanged by HU.

Page 19, line 19– page 20, line 5

The PINK1/Parkin pathway is affected by both S-nitrosylation and nitration [28,50]. The approximately 40 kDa Parkin detected by IP may be related to the acetone precipitation method used, as the insoluble protein was observed in the IP buffer containing sample proteins after acetone precipitation in both groups. Despite this, the S-nitrosylation and nitration of Parkin were unchanged by HU (Fig 5). These results indicate that Parkin is not inhibited by excessive S-nitrosylation and nitration after 14 days of HU. Additionally, HU did not alter the expression of PINK1 and Parkin (Fig 6). These results suggest that the PINK1/Parkin pathway may not be actively involved in mitophagy in atrophied soleus muscles after 14 days of HU. Previous studies showed that the expression of PINK1 and/or Parkin was not increased in the GAS muscles after muscle inactivity, suggesting that the PINK1/Parkin pathway does not play a major role in skeletal muscle inactivity-induced mitophagy [33-36]. Although the muscles examined in the present study differ from those evaluated in the previous studies [33-36], the PINK1/Parkin pathway may be inactive in atrophied soleus muscles after 14 days of HU. 

Page 20, line 6–18

Analysis of the expression and ubiquitination of MFN2 indicated that soleus muscles with HU-induced atrophy exhibited the high molecular weight MFN2 accumulation and decreased MFN2 ubiquitination (Fig 6 and 7). These findings suggest that the activity of ubiquitin ligases responsible for the ubiquitination of MFN2 is reduced in atrophied soleus muscles. Additionally, these results also support the possibility of the attenuated degradation of some proteins such as nNOS in atrophied muscles. MFN2 is ubiquitinated by various E3 ligases such as Parkin, mitochondrial E3 ubiquitin ligase 1, MARCH5/MITOL, HUWE1, and glycoprotein 78 [64]. We did not determine which pathway is related to the reduced ubiquitination of MFN2. However, a reduction in Parkin activity may contribute to the accumulation and reduced ubiquitination of MFN2 in atrophied soleus muscles. As Parkin is also an HSP70-binding E3 ubiquitin ligase [65], the absence of HSP70 inhibits the Parkin E3 ubiquitin ligase activity [31]. In addition, HSP72 knockout mice showed impaired Parkin activity and mitophagy in the skeletal muscle [66]. Thus, the basal activity of Parkin may be attenuated by reduced HSP70 expression in the atrophied soleus muscles induced by HU.

Conclusion section

Page 22, line 2–6

Moreover, the PINK1/Parkin pathway may not play a role in the mitophagy of atrophied soleus muscles at this time point. In contrast, the activity of Parkin may be downregulated because of reduced HSP70 expression, which may contribute to the accumulation and reduced ubiquitination of a mitochondrial fusion protein in atrophied soleus muscles after 14 days of HU.

Minor concerns

1. Page 3, line 3: needs a reference.

Response: Thank you for your suggestion. We have added a reference to support the sentence as follows:

Page 3, line 3–4.

Skeletal muscle atrophy occurs because of both a reduced rate of protein synthesis and an increased rate of degradation [1-3].

2. Page 14, line 9: “MFN2 immunoreactivity between 110 and 160 kDa” This signal might or might not be MFN2. Is there any prove from proteomics data / mass spectrometry that this is indeed MFN2?

Response: We agree with the reviewer’s opinion. Mass spectrometry in addition to western blotting and immunoprecipitation analysis supports certainty (credibility) of the results. However, in this study, mass spectrometry could not be performed because of a lack of samples. MFN2 immunoreactivity occured between 110 and 160 kDa is MFN2 because 1) the existence of high molecular weight MFN2 was demonstrated in previous studies, 2) high molecular weight MFN2 was detected using a slightly different analysis. Western blot analysis was performed to detect MFN2 in a lysate and by immunoprecipitation to enrich MFN2 to enable detection.

 We have explained the certainty of MFN2 immunoreactivity between 110 and 160 kDa in the Results section of the revised manuscript as follows:

In the results section

Page 16, line 1–6.

We additionally confirmed whether the high molecular weight MFN2 could be detected by IP. Fig 7A shows the immunoreactivities of MFN2 and ubiquitin after MFN2 IP. Immunoreactivity of MFN2 was observed between 110 and 160 kDa in both groups, which was similar to the results obtained for western blot analysis of MFN2 (Fig 6A). This suggests that the high molecular weight MFN2 is present in the atrophied soleus muscles and proves MFN2 immunoreactivity between 110 and 160 kDa (Fig 6A).

Page 16, line 15–16

Therefore, these results indicate that the ubiquitination of MFN2 is decreased in HU-induced soleus muscle atrophy.

3. Page 14, line 4 from bottom: “MEF2” is probably a typo.

Response: Thank you for pointing this out. This error has been corrected (page 15, line 9).

4. Please add indicators of molecular weight in all Western blots shown.

Response: Thank you for pointing out. We have added the molecular weight to all western blots.

5. Figure 7 is not conclusive. It is not sure if any MFN2 signal was detected in A (top panel) as MFN2 is supposed to migrate at 86 kDa. The anti-ubiquitin Blot is not convincing. The decrease in “Ubiquitinated MFN2” (B) appears to result from increased MFN2 160 kDa, which has not even been shown to be MFN2. This data set can be left out of the manuscript.

Response: Thank you for your suggestion. Non-ubiquitinated MFN2 can be detected at approximately 80-86 kDa, although there are some differences depending on the molecular weight marker used. We detected non-ubiquitinated MFN2 at approximately 80 kDa in immunoprecipitated samples. In addition, we confirmed whether the high molecular weight MFN2 could be detected by immunoprecipitation; we detected the immunoreactivity between 110 to 160 kDa. This result supports the certainty of MFN2 immunoreactivity between 110 and 160 kDa based on western blot data.

MFN2 is ubiquitinated at multiple sites (Escobar-Henriques et al., Front. Physiol. (2019) 10:517.). In addition, the binding of ubiquitin moieties to ubiquitinated MFN2 results in the formation of poly-ubiquitin chains. Ubiquitin is an 8.6 kDa protein. When MFN2 is poly-ubiquitinated at multiple sites, its molecular weight is increased. Therefore, high molecular weight MFN2 can be detected. 

The existence of high molecular weight MFN2 has already been shown in many previous studies in the context of MFN2 ubiquitination and have been used to evaluate MFN2 ubiquitination (Tanaka et al., J Cell Biol. 2010; 191(7): 1367-1380., Ziviani et al. Proc Natl Acad Sci U S A. 2010; 107(11): 5018-5023., Gegg et al., Human Molecular Genetics, 2010, Vol. 19, No. 24 4861–4870., Di Rita et al., Nature Communications (2018)9:3755, DOI: 10.1038/s41467-018-05722-3., Sugiura et al., Molecular Cell 2013; 51(1): 20-34., Chen et al., Am J Physiol Endocrinol Metab 315: E404–E415, 2018.). 

6. Page 10: “These results indicated that … the increase of nNOS expression was caused by the increased expression of p-nNOS at Ser1446.”. Direct comparison of the Western blot data indicates that there is not a direct correlation between nNOS and p-nNOS Ser1446 so that this conclusion is questionable. It rather appears that the increase in p-nNOS at Ser1446 was caused by an increase in nNOS contents.

Response: Thank you for your suggestion. We have rewritten the sentences as follows: 

In the Results section

Page 11, line 2–3

These results indicate nNOS expression and p-nNOS at Ser1446 were increased in response to 14 days of HU-induced atrophy in soleus muscles.

---

## [Decision Letter · Decision Letter 1]

25 Nov 2020

Potential roles of neuronal nitric oxide synthase and the PTEN-induced kinase 1 (PINK1)/Parkin pathway for mitochondrial protein degradation in disuse-induced soleus muscle atrophy in adult rats

PONE-D-20-25099R1

Dear Dr. Uda,

We’re pleased to inform you that your manuscript has been judged scientifically suitable for publication and will be formally accepted for publication once it meets all outstanding technical requirements.

Kind regards,

Michael Bader

Academic Editor

PLOS ONE

Additional Editor Comments (optional):

Reviewers' comments:

Reviewer's Responses to Questions

**Comments to the Author**

1. If the authors have adequately addressed your comments raised in a previous round of review and you feel that this manuscript is now acceptable for publication, you may indicate that here to bypass the “Comments to the Author” section, enter your conflict of interest statement in the “Confidential to Editor” section, and submit your "Accept" recommendation.

Reviewer #1: (No Response)

Reviewer #2: All comments have been addressed

2. Is the manuscript technically sound, and do the data support the conclusions?

Reviewer #1: Yes

Reviewer #2: Yes

3. Has the statistical analysis been performed appropriately and rigorously? 

Reviewer #1: Yes

Reviewer #2: Yes

4. Have the authors made all data underlying the findings in their manuscript fully available?

Reviewer #1: Yes

Reviewer #2: Yes

5. Is the manuscript presented in an intelligible fashion and written in standard English?

Reviewer #1: Yes

Reviewer #2: Yes

6. Review Comments to the Author

Reviewer #1: The authors have overall adequately responded to comments. Below are a few small additional comments/suggestions.

There are antibodies for p-Parkin at Ser56, one company is Abcam, catalog #ab154995

Since the authors did not add a post-hoc power estimate, I recommend adding a sentence or two in the limitations portion on the sample size of 4.

I would still recommend individual data points for the soleus data since the number of groups and sample size allows for this without “messy” graphs, but this is more of a personal preference than a necessity.

Reviewer #2: The authors have adressed all questions raised and revised the manuscript accordingly. I am still not convinced about ubiquitinated MFN2 and its quantification (Figure 7 A, B).

7. PLOS authors have the option to publish the peer review history of their article (what does this mean?). If published, this will include your full peer review and any attached files.

Reviewer #1: No

Reviewer #2: No

---

## [Editor Report · Acceptance letter]

1 Dec 2020

PONE-D-20-25099R1 

Potential roles of neuronal nitric oxide synthase and the PTEN-induced kinase 1 (PINK1)/Parkin pathway for mitochondrial protein degradation in disuse-induced soleus muscle atrophy in adult rats 

Dear Dr. Uda:

I'm pleased to inform you that your manuscript has been deemed suitable for publication in PLOS ONE. Congratulations! Your manuscript is now with our production department. 

Kind regards, 

on behalf of

Prof. Michael Bader 

Academic Editor

PLOS ONE